**Insights into carbonate environmental conditions in the Chukchi Sea**
Claudine Hauri[1], Brita Irving[1], Sam Dupont[2,3], Rémi Pagés[1], Donna D. W. Hauser[1], and Seth L.
Danielson[4]
[1] International Arctic Research Center, University of Alaska Fairbanks, Fairbanks, AK 99775,
USA
[2] Department of Biological and Environmental Sciences, University of Gothenburg,
Fiskebäckskil 45178, Sweden
[3] Radioecology Laboratory International Atomic Energy Agency (IAEA), Marine Laboratories,
Principality of Monaco
[4] College of Fisheries and Ocean Science, University of Alaska Fairbanks, Fairbanks, AK 99775,
USA
Correspondence email: chauri@alaska.edu
**Abstract**

17        Healthy Arctic marine ecosystems are essential to the food security and sovereignty, culture,

and wellbeing of Indigenous Peoples in the Arctic. At the same time, Arctic marine ecosystems
are highly susceptible to impacts of climate change and ocean acidification. While increasing
ocean and air temperatures and melting sea ice act as direct stressors on the ecosystem, they also
indirectly enhance ocean acidification, accelerating the associated changes in the inorganic
carbon system. Yet, much is to be learned about the current state and variability of the inorganic
carbon system in remote, high-latitude oceans. Here, we present time-series (2016-2020) of pH
and the partial pressure of carbon dioxide ($pCO_2$) from the northeast Chukchi Sea continental
shelf. The Chukchi Ecosystem Observatory includes a suite of subsurface year-round moorings
sited amid a biological hotspot that is characterized by high primary productivity and a rich
benthic food web that in turn supports coastal Iñupiat, whales, ice seals, walrus (*Odobenus*
*rosmarus*), and Arctic cod (*Boreogadus saida*). Our observations suggest that near-bottom
waters (33 m depth, 13 m above the seafloor) are a high carbon dioxide and low pH and
aragonite saturation state ($\Omega_{arag}$) environment in summer and fall, when organic material from the
highly productive summer remineralizes. During this time, $\Omega_{arag}$ can be as low as 0.4. In winter,
when the site was covered by sea ice, pH was < 8 and $\Omega_{arag}$ remained undersaturated under the
sea ice. There were only two short seasonal periods with relatively higher pH and $\Omega_{arag}$, which
we term ocean acidification relaxation events. In spring, high primary production from sea ice
algae and phytoplankton blooms led to spikes in pH (pH > 8) and aragonite oversaturation. In
late fall, strong wind driven mixing events that delivered low $CO_2$ surface water to the shelf also
led to events with elevated pH and $\Omega_{arag}$. Given the recent observations of high rates of ocean
acidification, and sudden and dramatic shift of the physical, biogeochemical, and ecosystem
conditions in the Chukchi Sea, it is possible that the observed extreme conditions at the Chukchi
Ecosystem Observatory are deviating from carbonate conditions to which many species are
adapted.
**1. Introduction**
The quickly changing Arctic Ocean has climatic, societal, and geopolitical implications for
the peoples of the Arctic and beyond (Huntington et al., 2022). Arctic Indigenous Peoples are at
the forefront of this change and their food security, food sovereignty, culture, and ways of life
depend on healthy Arctic marine ecosystems (ICC, 2015). The Arctic is warming at a rate that is
up to four times that of the rest of the globe (Serreze and Barry, 2011; Serreze and Francis, 2006;
Rantanen et al., 2022). This phenomenon, called Arctic Amplification, is observed in air and sea
temperatures, has accelerated in recent years, and is expected to continue in the future (Rantanen
et al., 2022; Shu et al., 2022). Warming exerts a toll on sea ice extent, ice thickness, and the
duration of seasonal sea ice cover: ice is forming later in fall and retreating earlier in spring,
thereby increasing the length of the open water period (Stroeve et al., 2011; Serreze et al., 2016;
Wood et al., 2015; Stroeve et al., 2014). The lowest Arctic wide minimum sea ice extents were
recorded during the last 16 years of the 44 year-long satellite time-series (National Snow and Ice
Data Center).
At the same time, the Arctic Ocean is vulnerable to ocean acidification. Although oceanic
uptake of anthropogenic carbon dioxide ($CO_2$) increases oceanic $CO_2$ and decreases pH and
calcium carbonate ($CaCO_3$) saturation states of calcite ($\Omega_{calc}$) and aragonite ($\Omega_{arag}$) globally,
climate induced changes to riverine input, temperature, sea ice, and circulation are accelerating
the rate of ocean acidification in the Arctic Ocean like nowhere else in the world (Woosley and
Millero, 2020; Qi et al., 2022a; Yamamoto-Kawai et al., 2009; Orr et al., 2022; Semiletov et al.,
2016; Qi et al., 2017). Recent observational studies propose that freshening of the Arctic Ocean
due to increased riverine input may play an even greater role in acidifying the Arctic Ocean than
the uptake of anthropogenic $CO_2$ (Woosley and Millero, 2020; Semiletov et al., 2016). In
addition, the cold Arctic waters have naturally low concentrations of carbonate ions ($CO_3^{2-}$) and
are therefore closer to aragonite undersaturation ($\Omega_{arag} < 1$) than more temperate waters (Orr,
2011; Sarmiento and Gruber, 2006), which leads to the chemical dissolution of free aragonitic
$CaCO_3$ structures (Bednaršek et al., 2021). Because of the naturally low concentrations of $CO_3^{2-}$,
such high latitude waters have a lower capacity to take up anthropogenic $CO_2$ and buffer these
changes (Orr, 2011). As a result, concentrations of hydrogen ions ($H^+$) increase and pH decreases
faster in the Arctic than in the tropics, for example.

In the Pacific Arctic, the Chukchi shelf waters have warmed by 0.45 $^{\circ}$C decade$^{-1}$ since 1990,

triple the rate since the beginning of the data record in 1922 (Danielson et al., 2020). Direct
observations of the inorganic carbon dynamics of the Chukchi Sea are mostly limited to June
through November because of the region's remoteness and accessibility during sea ice covered
months. Summertime profiles across the Chukchi Sea show steep vertical gradients in inorganic
carbon chemistry (Bates, 2015; Bates et al., 2009; Pipko et al., 2002; Mathis and Questel, 2013).
Surface waters have a low partial pressure of carbon dioxide ($p$CO$_2$) as a result of high primary
production after sea ice retreat, leading to aragonite supersaturated conditions, with $\Omega_{arag} > 2$
(Bates, 2015; Bates et al., 2009). In areas with sea ice melt or riverine freshwater influence, $\Omega_{arag}$
tends to be lower and at times undersaturated (Bates et al., 2009; Yamamoto-Kawai et al., 2009).
At the same time, $p$CO$_2$ values near the seafloor are around 1000 µatm as a result of organic
matter, leading to summertime aragonite undersaturation (Mathis and Questel, 2013; Pipko et al.,
2002; Bates, 2015). Between September and November, continuous measurements from within a
few meters of the surface suggest a mosaic of $p$CO$_2$ levels between ~ 200 to 600 μatm, likely due
to patchy wind-induced mixing entraining high-CO$_2$ waters from depth into the surface mixed
layer (Hauri et al., 2013). Yamamoto-Kawai et al. (2016) used mooring observations of S, T, and
apparent oxygen utilization to estimate dissolved inorganic carbon (DIC), total alkalinity (TA),
and $\Omega_{arag}$ in bottom waters at their mooring site in the Hope Valley in the southwestern Chukchi
Sea to give first insights into year round variability of the inorganic carbon system. They found
slightly less intense aragonite undersaturation in spring and winter compared to summer, with a
net undersaturation duration of 7.5-8.5 months per year.

The Chukchi Ecosystem Observatory (CEO) is situated in a benthic hotspot (Figure 1) where

high primary production supports rich and interconnected benthic and pelagic food webs
(Grebmeier et al., 2015; Moore and Stabeno, 2015). The benthos is dominated by calcifying
bivalves, polychaetes, amphipods, sipunculids, echinoderms and crustaceans (Grebmeier et al.,
2015; Blanchard et al., 2013). Benthic foraging bearded seals (*Erignathus barbatus*), walrus
(*Odobenus rosmarus divergens*), gray whale (*Eschrichtius robustus*), and seabirds feed on these
calcifiers during the open water season (Kuletz et al., 2015; Jay et al., 2012; Moore et al., 2022).
The CEO site, located on the southern flank of Hanna Shoal, is a region of reduced stratification
(relative to other sides of the shoal) that likely alternately feels the effects of differing flow
regimes located to the west and to the east (Fang et al., 2020). Consequently, the site exhibits
relatively weaker currents (Tian et al., 2021) and so is conducive to deposition of sinking organic
matter that in turn feeds the local benthos (Grebmeier et al., 2015). Prolonged open-water
seasons during periods of high solar irradiance, in combination with an influx of new nutrients
and wind mixing, are likely enhancing primary and secondary production as well as advection of
zooplankton (Lewis et al., 2020; Arrigo and van Dijken, 2015; Wood et al., 2015). These
physical processes in turn fuel keystone consumers such as Arctic cod (*Boreogadus saida*) and
upper trophic level ringed seals (*Phoca hispida*), beluga (*Delphinapterus leucas*) and bowhead
whales (*Balaena mysticetus*) as well as predatory polar bears (*Ursus arctos*) and Indigenous
People who rely on the marine ecosystem for traditional and customary harvesting (Huntington
et al., 2020).

Perturbation of the seawater carbonate system associated with ocean acidification and

climate change can have significant physiological and ecological consequences for marine
species and ecosystems (Doney et al., 2020). All parameters of the carbonate system (pH, $p$CO$_2$,
$\Omega_{arag}$, concentrations of HCO$_3^-$, CO$_3^{2-}$, etc.) have the potential to affect the physiology of marine
organisms while a change in the saturation state ($\Omega$) can lead to the dissolution of unprotected or
"free" CaCO$_3$ structures. Recent work has highlighted the importance of local adaptation to the
present environmental variability as a key factor driving species sensitivity to ocean acidification
(Vargas et al., 2017, 2022). As carbonate chemistry conditions vary enormously between
regions, marine organisms are naturally exposed to different selective pressures and can evolve
different strategies to cope with low pH or $\Omega$, or high $p$CO$_2$. For example, the deep-sea mussel
*Bathymodiolus brevior* living around vents at 1600 m depths is capable of precipitating calcium
carbonate at pH ranging between 5.36 and 7.30 and highly undersaturated waters (Tunnicliffe et
al., 2009). The response to changes in the carbonate chemistry is also modulated by other
environmental drivers such as temperature or food availability (e.g. Thomsen et al., 2013;
Breitberg et al., 2015). Consequently, no absolute or single threshold is expected for ocean
acidification (e.g., Bednaršek et al., 2021) and a pre-requisite to assessing the impact on any
biota is the monitoring at a short temporal scale to characterize the present environmental niche.
When it comes to future impacts, the more intense and faster the changes associated with ocean
acidification, the more adverse associated biological impacts are expected (Vargas et al. 2017,
2022). As a result, it is anticipated that Arctic marine waters that are experiencing widespread
and rapid ocean acidification will potentially undergo severe negative ecosystem impacts
(AMAP 2018).
Here, we present satellite sea ice coverage data and four years of nearly continuous salinity,
temperature, and $p$CO$_2$ data, accompanied by pH, nitrate (NO$_3$), dissolved oxygen (O$_2$), and
chlorophyll fluorescence data for some of the time (Table 1, Figures 2 and 3). We developed an
empirical equation for estimating pH from moored $p$CO$_2$, temperature, and salinity and evaluated
it using discrete samples collected across the Chukchi Sea, Bering Sea, and Beaufort Sea. Our
timeseries allow us to assess the seasonal and interannual variability and controls of the
inorganic carbon system in the Chukchi Sea between 2016 and 2020 and characterize the
chemical conditions experienced by organisms. We discuss our observations in terms of
progressing acidification and implications to organisms in the Chukchi Sea region.

**2.  Materials and Methods**
**2.1 The Chukchi Ecosystem Observatory (CEO)**
The Chukchi Sea is a shallow shelf sea with maximum depths < 50 m. It is largely a
unidirectional inflow shelf system with Pacific origin water entering the Chukchi Sea through the
Bering Strait and advecting north into the Arctic Ocean (Carmack and Wassmann, 2006). The
CEO (71°36' N, 161°30' W, Figure 1, Hauri et al., 2018) is located along the pathway of waters
flowing through Bering Strait (Fang et al., 2020) and thence from the west of Hanna Shoal
toward Barrow Canyon to the south, although the wind can also drive waters from the east over
the observatory site (Fang et al., 2020). From both shipboard and moored acoustic Doppler
current profiler records, the south side of Hanna Shoal mean flow is characterized by a weak
southward-directed current (Tian et al., 2021).

The observatory consists of oceanographic moorings that sample year-round, equipped with a

variety of sensors that measure sea ice cover and thickness (Sandy et al., 2022), light, currents,
waves, salinity, temperature, concentrations of dissolved oxygen, nitrate, and particulate matter,
pH, $pCO_2$, chlorophyll fluorescence, zooplankton abundance and vertical migration (Lalande et
al., 2021, 2020), the presence of Arctic cod and zooplankton (Gonzalez et al., 2021), and the
vocalizations of marine mammals. During some years, the observatory included a third mooring,
an experimental "freeze-up detection mooring", which transmitted real-time data of conductivity
and temperature throughout the water column until sea ice formation. The primary moorings
stretch from the seafloor at 46 m to about 33 m depth, designed to avoid collisions with ice keels.
Pressure sensors at the top of the moorings show less than ± 1 m of excursion of the moored
sensor package from its deployment mean depth in any given year, indicating that mooring blow-
over or diving is not the cause of any observed large variability. Description of the CEO and lists
of sensors deployed at the site can be found in Danielson et al. (2017) and Hauri et al., (2018).
For this study we focus on the inorganic carbon system and its controlling mechanisms.

**2.2 $pCO_2$**

We used a CONTROS HydroC $CO_2$ sensor (4H-Jena Engineering GmbH, Kiel, Germany) to

measure $pCO_2$. The Contros HydroC $CO_2$ sensor was outfitted with a pump (SBE 5M, Sea-Bird
Electronics) that flushes ambient seawater against a thin semi permeable membrane, which
serves as equilibrator for dissolved $CO_2$ between the ambient seawater and the headspace of the
sensor. Technical details about the sensor and its performance are described in Fietzek et al.
(2014), who estimated sensor accuracy to be better than 1% with postprocessing.

A HydroC $CO_2$ sensor has been deployed at the CEO site since 2016. In all deployments,

except 2016, HydroC $CO_2$ sensors were post-calibrated. The lack of post-calibration in 2016 is
not expected to negatively affect data quality because a battery failure resulted in data returns
only over the first 3 months (August through November). Following a zero interval where the
gas was pumped through a soda lime cartridge to create a zero-signal reference with respect to
$CO_2$, and subsequent flush interval to allow $CO_2$ concentrations to return to ambient conditions,
measurements were taken in a burst fashion every 12 or 24 hours depending on deployment year
(Table 1). Average $pCO_2$ values are reported as the mean of the measure interval (Table 1) with
standard uncertainty (Equation 1) defined following best practices (Orr et al., 2018) and where
the random component is the standard deviation of the mean, and the systematic components
include sensor accuracy and estimated error of the regression during calibration.
$$u = \sqrt{u^2_{systematic} + u^2_{random}} \qquad (1)$$
More than 96% of the time, the relative uncertainty of the $pCO_2$ data met the weather data
quality goal, defined as 2.5% by the Global Ocean Acidification Observing Network (GOA-ON,
Newton et al., 2015).

HydroC $CO_2$ data were processed using Jupyter notebook scripts developed by 4H-Jena

Engineering GmbH using pre- and post-calibration coefficients interpolated with any change in
the zero-signal reference over the deployment (Fietzek et al., 2014). Further processing using in-
house MATLAB scripts included removal of outliers, calculation of the average $pCO_2$, and
calculation of uncertainty estimates for each measure interval.

**2.3 pH**

A SeapHOx sensor (Satlantic SeaFET™ V1 pH sensor integrated with Sea-Bird Electronics SBE 37-SMP-ODO) was used to concurrently measure pH, salinity, temperature, pressure, and oxygen (Martz et al., 2010).  A SeapHOx was deployed at CEO in 2016, 2017, and 2018. No SeapHOx was deployed in 2019 or 2020 due to supply chain delays and communication issues at sea. Unfortunately, measured pH ($pH_{SeaFET}$) from the 2016 and 2018 SeapHOx deployments were unusable due to high levels of noise in both the internal and external electrodes. In short, we only have usable pH data between August 2017 and August 2018.

$pH_{SeaFET}$ data were excluded during a 14-day conditioning period following deployment and were processed with post-calibration corrected temperature and salinity from the SBE37 following Bresnahan et al. (2014) using voltage from the external electrode ($V_{ext}$), and $pH_{Vext}$ (pH calculated from the external electrode of the SeaFET) from an extended period of low variability (18 February 2018). Despite the availability of discrete data from one calibration cast (Cross et al., 2020b; Table 2), $pH_{Vex}$ was used as the single calibration point (Bresnahan et al., 2014) for a variety of reasons: 1) high variability of $pH_{SeaFET}$ (0.0581 pH units) straddling a 12 hour window around the discrete sample collection time, 2) high temporal and spatial variability often seen in the Chukchi Sea, and 3) the discrete pH sample was within the published SeaFET accuracy of 0.05 (Table 2, Figure S1). $pH_{SeaFET}$ values are reported as the mean of the measure interval (Table 1) and standard uncertainty is calculated with Equation 1 with the standard deviation of the average (random), and the SeaFET accuracy (systematic).  Data handling and processing were done using in-house MATLAB scripts. pH is reported in total scale and at *in situ* temperature for the entirety of this paper.

**2.4 Nitrate**
$NO_3$ measurements were from a Submersible Ultraviolet Nitrate Analyzer (SUNA) V2 by
Sea-Bird Scientific. The SUNA is an *in situ* ultraviolet spectrophotometer designed to measure
the concentration of nitrate ions in water. SUNA V2 data were processed using a publicly
available toolbox (Hennon et al., 2022; Irving, 2021) with QA/QC steps that included thermal
and salinity corrections (Sakamoto et al., 2009), assessment of spectra and outlier removal based
on spectral counts (Mordy et al., 2020), and concentration adjustments (absolute offset and linear
drift) based on pre-deployment and post-recovery reference measurements of zero concentration
(DI) water and a nitrate standard and, when available, nutrient samples taken from Niskin bottles
near the mooring site (e.g. Daniel et al., 2020).

**2.5 CTD and Oxygen**
Two CTDs were deployed on the CEO mooring near the HydroC $CO_2$ depth. The main
pumped Sea-Bird SeaCAT (SBE16) has been deployed on the CEO mooring around 33 m depth
since 2014. A pumped SBE43 oxygen sensor was deployed with the SBE16 during the 2015-
2016, 2017-2018, and 2019-2020 deployments but only data returns from the 2017-2018
deployment is discussed briefly in this manuscript (Figure S2).
The other pumped CTD was a Sea-Bird MicroCAT (SBE37-SMP-ODO), which was
integrated with an optical dissolved oxygen sensor (SBE63; Figure S2), and the SeaFET pH
sensor within the SeapHOx instrument. The SeapHOx was deployed in fall 2016, 2017, and
2018. The SBE37-SMP-ODO did not record any CTD or oxygen data during the 2016
deployment and only recorded CTD and oxygen data between August and November 3 in 2018
due to battery failure.

Processing of these data included temperature and conductivity correction using pre- and

post-calibration data following Sea-Bird Application Note 31 and oxygen correction using pre-

and post-calibration data following Sea-Bird Module 28. Oxygen was converted from ml/l to

249       µmol/kg following Bittig et al. (2018). Density and practical salinity were calculated using the

TEOS-10 GSW Oceanographic Toolbox (McDougall and Baker, 2011).

Differences between the two oxygen sensors (SBE43 and SBE63) of approximately 145 to

265 µmol/kg were observed over the 2017-2018 deployment, and both moored sensors had

varying offsets compared to nearby casts (Figure S2).  Therefore, only relative oxygen values

from the freshly calibrated SBE63 are discussed in this paper.

The freeze-up detection mooring (Figure 6) consisted of four Sea-Bird SBE 37 inductive

modem CTD sensors that transmitted in real time hourly temperature, salinity, and pressure data

via the surface float from four subsurface depths (8, 20, 30, and 40 m; Hauri et al., 2018).

**2.6 Development of empirical relationship to estimate pH**

Empirical relationships for estimating water column pH have been developed for regions

spanning southern, tropical, temperate and Arctic biomes, using a variety of commonly measured

parameters (e.g., pH(S, T, $NO_3$, $O_2$, Si) Carter et al 2018; pH($O_2$,T,S) Li et al., 2016; pH($\theta$,$O_2$)

Watanabe et al., 2020; pH($NO_3$, T, S, P) and pH($O_2$, T, S, P) Williams et al., 2016; pH($O_2$, T)

Alin et al., 2012; pH($O_2$, T) and pH($NO_3$, T) Juranek et al., 2009). Given the tight coupling

between the concentration of $H^+$ and concentration of $CO_2$ solution, an empirical relationship for

estimating surface pH from $p$$CO_2$ was developed by the National Academies of Sciences,

Engineering and Medicine (2017) appendix F. Licker et al. (2019) used this empirical

relationship to calculate the global average surface ocean pH and found it represented the

relationship for surface water temperatures spanning 5°C to 45°C. Here, we take a similar
approach but extend it to water column pH in our cold region using temperature (T) and salinity
(S) as additional proxy parameters (Equation 2).
$$pH^{est} = \alpha_0 + \alpha_1 \log(pCO_2) + \alpha_2 T + \alpha_3 S \qquad (2)$$
Where $pH^{est}$ is the estimated value of water column pH, $pCO_2$ is from the HydroC, and T and S
are from the SBE16, and all $\alpha$ ($\alpha_0$ = 10.4660, $\alpha_1$= -0.4088, $\alpha_2$ = 0.0013, $\alpha_3$ = -0.0001) terms are
model-estimated coefficients determined using MATLAB's multiple linear regression algorithm
*regress.m* (Chatterjee and Hadi, 1986). After interpolating $pH_{SeaFET}$ (Figure 4, red dots) to the
$pCO_2$ timestamp, the algorithm was trained over an arbitrarily chosen 180-day period
(15/9/2017-14/3/2018, Figure 4, dashed box). An uncertainty of 0.0525 for $pH^{est}$ (Figure 3 and
Figure S1, gray shading) was determined with Equation 1, where the RMSE (the uncertainty in
the estimation) over the entire $pH_{SeaFET}$ timeseries is the random component and the published
accuracy of the SeaFET is the systematic component (since the algorithm was trained with
$pH_{SeaFET}$). The algorithm cross-validation and evaluation are discussed in section 3.1. Unless
explicitly defined otherwise, observations of pH refer to $pH^{est}$ for the remainder of this paper.

**2.7 Carbonate system calculations**
Moored data were collected at different sample intervals (Table 1) and were linearly
interpolated to the HydroC $CO_2$ timestamp to enable further calculations.  TA, DIC, and $\Omega_{arag}$
(Figure 11 a & b and Figure 3d) were calculated based on measured $pCO_2$, S, T, and pressure (P)
and algorithm-based pH ($pH^{est}$). Due to a lack of data, nutrient concentrations (Si, $PO_4$, $NH_4$,
$H_2S$) were assumed to be negligible in the CO2SYS calculations (e.g. deGrandpre et al., 2019;
Vergara-Jara et al., 2019; Islam et al., 2017). $pH^{est}$ was used in lieu of $pH_{SeaFET}$ to allow for
calculations over the whole $p$CO$_2$ record and due to erroneously large variability of DIC and TA
when pH$_{SeaFET}$ was used as an input parameter (Raimondi et al., 2019; Cullison-Gray et al.,
2011). The pH-$p$CO$_2$ input pair leads to large, calculated errors in DIC and TA (Raimondi et al.,
2019; Cullison-Gray et al., 2011) due to strong covariance between the two parameters (both
temperature and pressure dependent). Cullison-Gray et al. (2011) attributed unreasonably large
short-term variability in calculated TA and DIC to temporal or spatial measurement mismatches
between input pH and $p$CO$_2$ parameters and found that appropriate filtering alleviated noise
spikes. By using pH$^{est}$, which by the nature of its definition is well correlated to $p$CO$_2$, we are
eliminating some of these spurious noise spikes. We show $\Omega_{arag}$ calculated from pH$_{SeaFET}$-$p$CO$_2$
(Figure 3d, red line) because it is less sensitive to calculated errors as it accounts for a small
portion of the total CO$_2$ in seawater (Cullison-Gray et al., 2011).
All inorganic carbon parameters were calculated using CO2SYSv3 (Sharp et al., 2023; Lewis
and Wallace, 1998) with dissociation constants for carbonic acid of Lueker et al. (2000),
bisulfate of Dickson (1990), hydrofluoric acid of Perez and Fraga (1987), and the boron-to-
chlorinity ratio of Lee et al. (2010). Sulpis et al. (2020) found that the carbonic acid dissociation
constants of Lueker et al. (2000) may underestimate $p$CO$_2$ in cold regions (below ~8°C), and
therefore overestimate pH and CO$_3^{2-}$. However, we choose to use Lueker et al. (2000) because
they are recommended (Dickson et al., 2007; Woosley, 2021), continue to be the standard (Jiang
et al., 2021; Lauvset et al., 2021), and are commonly used at high latitudes (Duke et al., 2021;
Raimondi et al., 2019; Woosley et al., 2017). Furthermore, the difference between DIC
calculated from pH$^{est}$ and $p$CO$_2$ and discrete samples interpolated to moored instrument depth
ranged from 266 to -195 μmol/kg using the k1 k2 of Sulpis et al. (2020), compared to -38 to -7
μmol/kg using Lueker et al. (2000).

**2.8 Sea ice concentration**

Sea ice concentration at the observatory site was taken from the National Snow and Ice Data

Center (NSIDC; DiGirolamo et al., 2022). Latitude and longitude coordinates were converted to

NSIDC's EASE grid coordinate system (Brodzik and Knowles, 2002) and the 25-km gridded

data were bilinearly interpolated to calculate sea ice concentration at the CEO site. Low sea ice is

defined by < 15 % sea ice coverage per grid cell.

**2.9 Estimation of model-based ocean acidification trend**

Model results were obtained from historical simulations of five different global Earth System

Models: 1) GFDL-CM4 (Silvers et al., 2018), 2) GFDL-ESM4 (Horowitz et al., 2018), 3) IPSL-

CM6A-LR-INCA (Boucher et al., 2020), 4) CNRM-ESM2-1 (Seferian, 2019), and 5) Max Plank

Earth System Model 1.2 (MPI-ESM1-2-LR, Wieners et al., 2019) that are part of the Coupled

Model Intercomparison Project Phase 6 (CMIP6). Each simulation was used to calculate the

annual trend of aragonite saturation state and pH at the closest depth and grid cell to the CEO

mooring.

**3. Results**

In the following, we will evaluate the pH algorithm (section 3.1), analyze the large

variability patterns (sections 3.2 and 3.3), and then take a closer look at the data from 2020 since

the seasonal cycle was different in 2020 than in previous years (section 3.4).

**3.1 pH algorithm**

The algorithm estimated pH data from the CEO site reasonably well and within the weather

uncertainty goal as defined by Newton et al. (2015) most of the time. As a first step, $pH^{est}$
consistency was assessed through cross-validation (Figure 5) using the test dataset (outside the
training period, $r^2 = 0.9666$, RMSE = 0.166) and across the whole timeseries ($r^2 = 0.9598$, RMSE
= 0.0161, p<0.0001, Figure 5). Observed high frequency spikes in $pH_{SeaFET}$ (Figure 4, red dots;
Figure 5d, red line) were not captured by the HydroC $pCO_2$ sensor (sampling frequency of 12 h)
and as a result, are not reproduced in the $pH^{est}$ timeseries. Throughout the $pH_{SeaFET}$ timeseries,
$pH^{est}$ overestimates $pH_{SeaFET}$ by a mean of 0.0008 and median of 0.0039. Since $pH^{est}$ generally
overestimates $pH_{SeaFET}$ (mean difference of 0.0008), we assume that $\Omega_{arag}$ is also somewhat
overestimated throughout this manuscript. Discrete water samples were used as reference values
to evaluate the algorithm at the CEO site (Table 2) and were found to be within the $pH^{est}$
uncertainty (Figure S1).

An independent verification of our algorithm was done using discrete data collected from the

Bering Sea to the Arctic Ocean on four research cruises in 2020, 2019, 2018, and 2017 (Figure
6d; Monacci et al., 2022; Cross et al., 2021; 2020a; 2020b), henceforth called the DBO dataset.
Samples collected from deeper than 500 m below the surface or flagged as questionable or bad
were excluded from this analysis. pH and $pCO_2$ were calculated from 1275 discrete samples
analyzed for TA, DIC, silicate, phosphate, and ammonium (except when silicate, phosphate, and
ammonium were assumed to be negligible for the 327 samples from SKQ202014S; Monacci et
al., 2022) using CO2SYSv3 (Sharp et al., 2023; section 2.7 for details) and are referred to as
$pH^{disc}_{calc}$ and $pCO_2^{disc}_{calc}$, respectively. $pH^{disc}_{est}$ was based on discrete water samples and
calculated using Equation 2 and was fit to $pH^{disc}_{calc}$ using a linear regression ($r^2 = 0.9975$, RMSE
= 0.0078, p-value < 0.0001; Figure 6 a – c). Mean and median differences between $pH^{disc}_{calc}$ and
$pH^{disc}_{est}$ were zero and 0.0022, respectively, with largest anomalies observed at lower salinities
(Figure 6c). Absolute differences between $pH^{disc}_{est}$ and $pH^{disc}_{cal}$ over the salinity range observed
at the CEO site (30.87 to 33.93) fall within the weather data quality goal (Newton et al., 2015)
98.7% of the time with maximum absolute differences < 0.03.  The uncertainty of 0.0154 for
$pH^{disc}_{est}$ was determined using Equation 1, where the mean combined standard uncertainty ($u_c$)
for $pH^{disc}_{calc}$ (0.0133; Orr et al., 2018) was the systmetic component, and the regression RMSE
was the random component.
Empirical relationships for estimating water column pH that rely on dissolved oxygen often
ignore surface waters to limit biases due to decoupling the stoichiometry of the $O_2$:$CO_2$
relationship due to air-sea gas exchange (e.g. Juranek et al., 2011; Alin et al., 2012; Li et al.,
2016). We see evidence of this bias in our algorithm at low salinity (Figure 6c) and low $pCO_2$
(not shown) when compared with the DBO dataset samples collected across the Arctic and from
the surface to 500 m, with $pH^{disc}_{est}$ overestimating $pH^{disc}_{calc}$ by a maximum of 0.049. If depth is
restricted to between 30 and 500 m when evaluating the algorithm with the DBO dataset,
algorithm performance improves ($r^2 = 0.9990$, RMSE = 0.0055, p-value < 0.0001; not shown)
and the maximum $pH^{disc}_{est}$ overestimates $pH^{disc}_{calc}$ by 0.022.

**3.2 Relaxation events**
The sub-surface waters at the CEO site comprise a high $pCO_2$, low pH, and low $\Omega_{arag}$
environment, with mean values of $pCO_2^{mean} = 538 \pm 7$ µatm, $pH^{mean} = 7.91 \pm 0.05$, $\Omega_{arag}^{mean} =$
$0.94 \pm 0.23$ across the full data record (Figure 3 b - d). In the following we will focus on spikes
of high pH and $\Omega_{arag}$ and low $pCO_2$ that occur in spring (May-June) and fall (September-
December); we define these spikes as relaxation events (see discussion for justification of term).
*Spring:* Springtime relaxation events at 33 m depth that exhibit relatively higher pH and
$\Omega_{arag}$ and lower $pCO_2$ compared to the overall mean, are likely consequences of photosynthetic
activity during sea ice break-up (Figures 2 and 3). In June of 2019 and 2020, near bottom pH and
$\Omega_{arag}$ spiked to > 8.17 and > 1.5, respectively, while $pCO_2$ dropped to < 286 µatm. $\Omega_{arag}$ remained
oversaturated and pH was greater than 8.0 for nearly all of June in 2018.  In 2019, the relaxation
event was less sustained, with only four short (2-6 day-long) events of relatively higher pH and
$\Omega_{arag}$ > 1 in June. In both years, chlorophyll fluorescence spiked and either $O_2$ increased (in
2018) or $NO_3$ decreased (in 2019), which are signs of photosynthetic activity and primary
production.
*Fall:* The relaxation events in fall were characterized by large and sudden drops in $pCO_2$,
abrupt increases in pH and $\Omega_{arag}$, and considerable interannual variability in their timing. Unlike
the relaxation events observed in spring, we attribute these fall relaxation events to wind-induced
physical mixing. To examine the controlling mechanisms causing these abrupt relaxation events
in fall, we will start with using water column salinity and temperature data from a freeze-up
detection buoy (Hauri et al., 2018) that was deployed in summer 2017 approximately 1 km away
from the biogeochemical mooring. The freeze-up detection mooring provided temperature and
salinity measurements every 7 meters throughout the water column from the time of its
deployment in mid-August until freeze-up. Data from the freeze-up detection mooring suggest
that warmer and fresher water from the upper water column gets periodically entrained down to
the location of the biogeochemical sensor package at 33 m depth, leading to enhanced variability
of density in August and September (Figure 7). Fluctuations of the pycnocline associated with
the passage of internal waves could also elevate signal variances. During this time $pCO_2$ often
decreased to or below atmospheric levels and pH sporadically reached values > 8. At the end of
September, a strong mixing event (with coincident strong surface winds) homogenized the water
column from the surface down to the location of the sensor package and caused a sudden
temperature increase from 0.4 °C to 3.9 °C (Figure 7c and 8a). At the same time, $p$CO$_2$ (Figure
7b and 8) decreased from 590 to 308 µatm. This suggests that warm and low CO$_2$ surface water
mixed with CO$_2$-rich subsurface water and led to a sustained relaxation period that subsequently
lasted until mid-November. Another mixing event further eroded the water column stratification
and replaced subsurface water with colder and fresher water (ice melt) from the surface at the
end of October. This second large mixing event did not lead to large changes in $p$CO$_2$, pH, and
$\Omega_{arag}$.

Salinity and temperature records from the biogeochemical mooring at 33 m depth also

suggest fall season mixing events in all other years, when increases in temperature coincide with
decreases in $p$CO$_2$ (Figure 8). For example, two mixing events shaped the carbonate chemistry
evolution in fall 2018. $p$CO$_2$ decreased from 915 µatm to around 565 µatm and $\Omega_{arag}$ increased to
0.9 as temperature increased and salinity decreased in early September (Figures 2 and 8). $p$CO$_2$
then increased to 1160 µatm in late October, before decreasing to 385 µatm at the beginning of
November, causing a spike in $\Omega_{arag}$ to 1.34. At the same time, salinity decreased by 1 unit,
suggesting a strong mixing event. Throughout November 2018, $p$CO$_2$ oscillated between 344 and
757 µatm and salinity between 31.01 and 32.97, hinting at additional mixing.

Similarly, an early mixing event in 2019 decreased $p$CO$_2$ to 352 µatm at the beginning of

September. Short-term variability in $p$CO$_2$ with maximum levels of up to 855 µatm and
minimum values below 300 µatm, variable temperature and salinity, and sporadic aragonite
oversaturation events point to mixing through mid-September. At the end of October, a large
mixing event homogenized the water column, accompanied by a decline of salinity by >1 unit,
increase of temperature to 4 °C, and decrease of $p$CO$_2$ from 565 µatm to below 400 µatm. In a
similar fashion to 2018, this fall mixing event was followed by a month-long period of large
variability of $p$CO$_2$, salinity, pH, and $\Omega_{arag}$, leading to short and sporadic aragonite oversaturation
events in November, and sustained oversaturation in December.

**3.3 Sustained periods of low pH and $\Omega_{arag}$, and high $p$CO$_2$**

Waters at 33 m depth at the CEO site were most acidified during the sea ice free periods

until mixing events entrained surface waters to the sensor depth (section 3.2). pH and $\Omega_{arag}$
started to gradually decrease from their maximum levels ($\Omega_{arag\_max}$ = 1.65, pH$_{max}$ = 8.19) at the
beginning of June in 2018 to their annual low at the beginning of November ($\Omega_{arag\_min}$ = 0.47,
pH$_{min}$ = 7.58, Figure 3 d and c). In November, the waters were also undersaturated with regards
to calcite (not shown) and $p$CO$_2$ peaked at 1159 µatm (Figure 3b). Dissolved oxygen decreased
by about 400 µmol kg$^{-1}$ between July and October, when the sensor stopped working properly.
The decrease of dissolved oxygen suggests remineralization of organic material. The decrease of
pH, $\Omega_{arag}$, O$_2$ and increase of $p$CO$_2$ was briefly interrupted by a strong mixing event in
September, which entrained warmer, fresher, and CO$_2$-poorer water down to 33 m depth (section
3.2, Figure 8). The 2019 observations paint a similar picture of remineralization during the
summer months, as the $p$CO$_2$ increase and pH and $\Omega_{arag}$ decreases were accompanied by an NO$_3$
increase.

$p$CO$_2$ steadily increased and pH and $\Omega_{arag}$ decreased during the sea ice covered periods

(Figures 8). pH was < 8 and $\Omega_{arag}$ remained undersaturated under the sea ice. At the same time,
NO$_3$ slowly increased and O$_2$ decreased, which points to slow organic matter remineralization
(Figure 9). Short-term variability in $p$CO$_2$, especially in January of all three observed years, was
also reflected in salinity, $O_2$ and $NO_3$ (Figure 9) and could be attributed to advection, as the CEO
site is adjacent to contrasting regimes of flow and hydrographic properties (Fang et al., 2020).

**3.4 Spring and summer of 2020 were different**
The seasonal cycle in 2020 strongly contrasted with the previous observed years. $pCO_2$
gradually increased by roughly 200 µatm throughout the sea ice covered months to 650 µatm
when sea ice started to retreat at the beginning of July. By the end of July, $pCO_2$ doubled and
increased to 1389 µatm, which is the highest $pCO_2$ level recorded in this timeseries. The peak of
$pCO_2$ was accompanied by an increase in salinity of 0.5 while temperature did not change,
suggesting the influence of advection. At the beginning of August, $pCO_2$ dropped to 536 µatm
and then oscillated around 600 µatm through much of August before returning to around 900
µatm for the next month. Similarly, pH decreased to 7.5 at the end of July and then oscillated
around 7.85, while $\Omega_{arag}$ dropped to 0.37, and oscillated around 0.85. The steep drop and
oscillation of $pCO_2$ was reflected in $NO_3$, suggesting that primary production and
remineralization played a role. When $pCO_2$ and $NO_3$ decreased at the beginning of August,
temperature simultaneously increased by 0.7 °C and salinity decreased by 0.12, suggesting that
entrainment of shallower water masses may have played a role too. Comprehensive analyses of
the factors that resulted in the 2020 differing conditions are beyond the scope of this paper, but
deserve attention in a future effort.

**4. Discussion**
CEO data provide new insights into the synoptic, seasonal and interannual variability of
the inorganic carbon system in a time when ocean acidification and climate change have already
started to transform this area. The observations suggest that the CEO site is a high-$CO_2$ and low-
pH and low-$\Omega_{arag}$ environment most of the time, except during sea ice break-up when the effects
of photosynthetic activity remove $CO_2$ from the system, and later in fall, when strong storm
events entrain low $pCO_2$ surface waters to the seafloor. Lowest pH and $CaCO_3$ saturation states
and highest $pCO_2$ occur in summer through late fall when organic matter remineralization
dominates the carbonate system balance. During this time, $\Omega_{arag}$ can fall below 0.5 and even $\Omega_{calc}$
becomes sporadically undersaturated ($\Omega_{calc} < 1$).

**4.1 pH algorithm**

Deploying oceanographic equipment in remote Arctic locations is challenging. The data

return from the SeapHOx sensors was disappointingly minimal, despite annual servicing and
calibration by the manufacturer. Our new pH algorithm is therefore even more important as it
fills pH data gaps in the CEO timeseries and can be applied with confidence from the Bering to
the Beaufort seas (Figure 6). While another successful year of moored pH data return at the CEO
site is needed to fully evaluate our algorithm throughout the year, comparison with single
discrete water samples nearby the CEO site and the DBO dataset (section 3.1, Table 2, Figures 6
and S1) suggest that our algorithm-derived pH meets the weather quality uncertainty goal of $\pm$
0.02 (Newton et al., 2015) much of the time.

The combination of our new algorithm with recent progress in monitoring $pCO_2$ with

Seagliders (Hayes et al., 2022) will further increase our ability to study the inorganic carbon
dynamics at times and locations when shipboard or mooring based measurements may not be
practical. Additional assessment is needed to determine to what degree the algorithm needs
adjustments beyond the region evaluated in this work.

## 4.2 Uncertainty

Inherent spatial and temporal variability of the inorganic carbon parameters in the
Chukchi Sea make the use of discrete water samples for evaluating sensor-based measurements
difficult. Historic continuous surface measurements from the area suggest that surface $pCO_2$ can
be as low $< 250$ µatm in early fall (Hauri et al., 2013), at a time of year when subsurface $pCO_2$
reaches its max of $>800$ µatm at the CEO site. This suggests a steep $pCO_2$ gradient of $> 17$ µatm
per meter. High-resolution pH data from the 2017/2018 deployment suggests high temporal
variability as well, further complicating the collection of discrete water samples to adequately
evaluate the sensors. The HydroC's zeroing function, in addition to our pre- and post-calibration
routines that factor into the post-processing of the data, gives us confidence in the accuracy of
the $pCO_2$ data, and further confidence in pH derived from $pCO_2$.
The $pH^{est}$ uncertainty of 0.0525 is likely a conservative estimate based on our validation
of $pH^{est}$ (section 3.1, Table 2). Consequently, propagated uncertainties in the calculated
parameters are high. As discussed in section 2.7, the pH-$pCO_2$ input pair exacerbates these larger
uncertainties. Mean TA($pH^{est}$,$pCO_2$), DIC($pH^{est}$,$pCO_2$), and $\Omega_{arag}$($pH^{est}$,$pCO_2$), $\pm$ u$_c$ (Orr et al.,
2018) are $2173 \pm 281$ µmol kg$^{-1}$, $2111 \pm 263$ µmol kg$^{-1}$, and $0.94 \pm 0.23$, respectively, when
input uncertainties are the standard uncertainty (Equation 1). When the input uncertainty for
$pH^{est}$ is only the RMSE of 0.0161 (section 3.1), uncertainties decrease to $\pm 98$ µmol kg$^{-1}$, $\pm 93$
µmol kg$^{-1}$, and $\pm 0.09$, respectively. When input uncertainties are only the random component of
the input parameters (i.e. standard deviation for $pH_{SeaFET}$ and $pCO_2$ and instrument precision for
T and S), TA($pH_{SeaFET}$,$pCO_2$), DIC($pH_{SeaFET}$,$pCO_2$), and $\Omega_{arag}$($pH_{SeaFET}$,$pCO_2$) u$_c$ drops to $\pm 38$
µmol kg$^{-1}$, $\pm 37$ µmol kg$^{-1}$, and $\pm 0.06$, respectively. Given the above uncertainties and that we
do not see significant biofouling at the CEO site, we believe that short term variability can be
discussed with confidence with this dataset. In other words, wiggles in the data represent real
events, despite the high uncertainty in the precise value of the calculated parameters.

**4.3 Subsurface biogeochemical drivers of pH, $\Omega_{arag}$, and $pCO_2$**
Inorganic carbon chemistry can be influenced by advection and vertical entrainment of
different water masses, temperature, salinity, biogeochemistry, and conservative mixing with TA
and DIC freshwater endmembers. Here, we followed Rheuban et al. (2019) and separated the
drivers of the observed large pH, $\Omega_{arag}$, and $pCO_2$ variability to provide additional insights into
our timeseries (Figure 10) using CO2SYS by altering input parameters temperature, salinity, TA,
and DIC. Anomalies relative to the reference values pH($T_0$, $S_0$, $DIC_0$, $TA_0$), $\Omega_{arag}$($T_0$, $S_0$, $DIC_0$,
$TA_0$), and $pCO_2$($T_0$, $S_0$, $DIC_0$, $TA_0$), were calculated using a linear Taylor series decomposition,
adding up the thermodynamic effects of temperature and salinity, and the perturbations due to
biogeochemistry, and conservative mixing with freshwater DIC and TA endmembers. (Rheuban
et al., 2019). Reference values $T_0$, $S_0$, $DIC_0$, and $TA_0$, are the mean of the CEO timeseries.
Freshwater from sea ice melt and meteoric sources (precipitation and rivers) may influence the
CEO site. TA and DIC concentrations of 450 $\mu$mol kg$^{-1}$ and 400 $\mu$mol kg$^{-1}$, respectively, have
been measured in Arctic sea ice (Rysgaard et al., 2007). Riverine input along the Gulf of Alaska
tends to have lower TA (366 $\mu$mol kg$^{-1}$) and DIC (397 $\mu$mol kg$^{-1}$) concentrations (Stackpoole et
al., 2016, 2017) than rivers draining into the Bering, Chukchi, and Beaufort Seas (TA = 1860
$\mu$mol kg$^{-1}$, DIC = 2010 $\mu$mol kg$^{-1}$, Holmes et al., 2021) all of which can influence the CEO site
to some extent (Asahara et al., 2012; Jung et al., 2021). In this Taylor decomposition we used sea
ice TA and DIC endmembers (Rysgaard et al., 2007) but want to emphasize that using Arctic
river endmembers did not meaningfully change the results (not shown). Figure 10 shows the
effects of biogeochemical processes, temperature, salinity, and conservative mixing with TA and
DIC freshwater endmembers on pH, $\Omega_{arag}$, and $pCO_2$. The effects of salinity (red) and
conservative mixing with TA and DIC freshwater endmembers (green) are negligible for pH,
$\Omega_{arag}$, and $pCO_2$. Temperature varied between -1.7 °C during the sea ice covered months and up
to 4 °C in late fall, when wind events mixed the whole water column and entrained warm and
low $pCO_2$ surface waters to the instrument depth at 33 m (see section 3.2 for a more in-depth
discussion of these mixing events). During this time, the increase in temperature counteracted the
effect of biogeochemistry slightly and increased $pCO_2$ and decreased pH (Figure 10 a,c).
Temperature did not affect $\Omega_{arag}$.

Biogeochemistry (photosynthesis, respiration, calcification, dissolution) is the most

important driver of the inorganic carbon dynamics at 33 m depth at the CEO site. The springtime
relaxation events in 2018 and 2019 with relatively higher pH and $\Omega_{arag}$, and lower $pCO_2$, were
mainly driven by biogeochemistry (Figure 10). During these events $O_2$ increased and $NO_3$
decreased, suggesting photosynthetic activity (Figure 2d, e and 3a). Near bottom photosynthetic
activity by phytoplankton or sea ice algae has been observed at different locations across the
Chukchi Sea (Arrigo et al., 2017; Ouyang et al., 2022; Stabeno et al., 2020; Koch et al., 2020).
Sediment trap data from a CEO deployment prior to the start of this $pCO_2$ and pH time-series
suggest that export of the exclusively sympagic sea ice algae *Nitzschia frigida* peaked in May
and June, during snow and ice melt events (Lalande et al., 2020), further supporting the
hypothesis that sea ice algae contributed to the $CO_2$ draw down. Interestingly, TA also increased
significantly during these events in 2018 and 2019, which cannot be solely attributed to organic
matter production. Specifically, TA increased by 23 umol kg$^{-1}$ in 2019 (Figure 11a). However,
with an observed $NO_3$ increase of 7.6 umol kg$^{-1}$, we would expect an increase of TA by 7.6 umol
kg-1. This is assuming that $NO_3$ is the primary source of nitrogen during organic matter
formation, and that assimilation of 1 umol of $NO_3$ leads to an increase of TA of 1 umol (Wolf-
Gladrow et al., 2007). The TA increase of 23 umol kg$^{-1}$ is therefore larger than expected from
organic matter formation alone and is likely due to $CaCO_3$ mineral dissolution. While direct
evidence is missing, the strong TA increase suggests that $CaCO_3$ mineral dissolution during sea
ice break up also plays an important role at the CEO site. As observed in other Arctic areas, it is
possible that ikaite crystals that were trapped in the ice matrix dissolved in the water column
when sea ice melted (Rysgaard et al., 2012, 2007).

**4.4 Progression of ocean acidification in the Chukchi Sea**
The Arctic Ocean acidification rate will continue to exceed the rate of $CO_2$ change in the
atmosphere because of the impacts of freshening and other more localized, seasonal or short-
term consequences of climate change (Woosley and Millero, 2020; Terhaar et al., 2021; Orr et
al., 2022; Qi et al., 2017). Seventeen years of ship-based data from sub surface Chukchi Summer
water suggests a mean pH change of -0.0047 ± 0.0026 and mean $\Omega_{arag}$ change of -0.017 ± 0.009
(Qi et al., 2022b). As a comparison, an average across historic simulations from five CMIP6
models (see methods) estimates a change in pH of -0.0077 year$^{-1}$ and $\Omega_{arag}$ of -0.0063 year$^{-1}$ at 33
m of the CEO site between 2002 – 2014. The historic CMIP6 simulations end in 2014 and
therefore miss the last years of extreme sea ice loss. Both observations and global model-based
trend estimates must be used with caution. The observations were collected during the sea ice
free period (Qi et al., 2022b), and therefore do not depict an annually representative trend.
Global models do not resolve important local physical, chemical, and biological meso-scale
processes and therefore mask out the variability of the inorganic carbon system and effects of
climate change.

Organisms living at the CEO site may have always been exposed to large seasonal

variability and low pH and $\Omega_{arag}$ (high $pCO_2$), but the combined and cumulative effects of
climate change and ocean acidification have rapidly made these conditions more extreme and
longer lasting. Ocean acidification serves as a gradual environmental press by increasing the
system's mean and extreme $pCO_2$ and decreasing mean and extreme pH and $\Omega_{arag}$. Climate
induced changes to other important controls of the inorganic carbon system, such as sea ice,
riverine input, temperature, and circulation can act as sudden pulses and further modulate the
inorganic carbon system to a less predictable degree and cause extreme events (Woosley and
Millero, 2020; Orr et al., 2022; Hauri et al., 2021; Qi et al., 2017). Huntington et al. (2020)
describe a sudden and dramatic shift of the physical, biogeochemical and ecosystem conditions
in the Chukchi and Northern Bering seas in 2017. For example, satellite data for the CEO site
illustrate that the longest open water seasons on record occurred between 2017 and 2020. Before
2017, the open water season was on average 81 ($\pm$ 40) days long (i.e., below 15 %
concentration), of which 60 ($\pm$ 44) days were ice free, whereas between 2017 and 2020, the low
sea ice period was 157 ($\pm$ 30) days long, of which 152 ($\pm$ 24) days were ice free (Figure 12). Sea
ice decline and increased nutrient influx has also promoted increased phytoplankton primary
production in the area (Lewis et al., 2020; Arrigo and van Dijken, 2015; Payne et al., 2021).
Since our inorganic carbon timeseries started after the "dramatic shift" that was observed in the
Chukchi Sea in 2017 (Huntington et al., 2020) and given the uncertainty in model output in this
region, we can only speculate about how the changes in sea ice, temperature and biological
production may have affected seasonal variability and extremes of the inorganic carbon
chemistry at the CEO site. However, since the summertime low pH and $\Omega_{arag}$ and high $p\text{CO}_2$ are
tightly coupled to the length of the ice-free period and intensity of organic matter production, it
is possible that the observed summertime period of extreme conditions may have been
previously unexperienced at this site. We therefore think it is justified to call the spikes of pH
and $\Omega_{arag}$ "ocean acidification relaxation events", since the long-lasting summertime period of
extremely low pH and $\Omega_{arag}$ may be a new pattern.

**4.5 Relevance for ecosystem**

Marine organisms are exposed to a wide range of naturally fluctuating environmental

conditions such as temperature, salinity, carbonate chemistry and food concentrations that
together constitute their ecological niche. As evolution works toward adaptation, the tolerance
range of species and ecosystems to such parameters varies between locations and is often closely
related to niche status (Vargas et al., 2022). Stress can be defined as a condition evoked in an
organism by one or more environmental and biological factors that bring the organism near or
over the limits of its ecological niche (after Van Straalen, 2003). The consequence of the
exposure to a stressor will depend on organismal sensitivity, stress intensity (how much it
deviates from present conditions) and stress duration. In a synthesis of the global literature on the
biological impacts of ocean acidification, Vargas et al. (2017, 2022) showed that the extreme of
the present range of variability of carbonate chemistry is a good predictor of species sensitivity.
In other words, larger deviations from present extreme high $p\text{CO}_2$ or extreme low pH, would be
expected to exert more negative biological impacts. Organismal stress and niche boundaries have
implications for the definition and understanding of controls and future ocean acidification
conditions in experiments aimed at evaluating future biological impacts.

637  Our data provide insights on conditions that affect and determine local species'

638 ecological niches, and a necessary key is to evaluate or re-evaluate their sensitivity to present and

639 future carbonate chemistry conditions, particularly for the sessile benthic calcifiers that constitute

640 prey for mobile and upper trophic level taxa. For example, an experimental study on three

641 common Arctic bivalve species (*Macoma calcarean, Astarte montagui* and *Astarte borealis*)

642 collected in the CEO concluded that these species were generally resilient to decreasing pH

643 (Goethel et al., 2017). However, only two pH were compared (a "control" (pH of 8.1) and an

644 "acidified" treatment (pH of 7.8) and our results show that organisms are already experiencing

645 more extreme conditions today than have been experimentally manipulated. While these data

646 provide insights on these species' plasticity to present pH conditions, they cannot be used to infer

647 sensitivity to future ocean acidification or extremes of current conditions. Based on the local

648 adaptation hypothesis (Vargas et al. 2017, 2022), stress and associated negative effect on species

649 fitness can be expected when pH deviates from the extreme of the present range of variability

650 (pH<7.5) as shown in other regions (e.g. echinoderms: Dorey et al., (2013); crustaceans: Thor

651 and Dupont, (2015); bivalves: Ventura et al., (2016)).

652  At the CEO, our results show sustained periods of remarkably low pH (e.g., 7.5; summer

653 to fall, winter). Higher pH values are observed in spring and late fall. While we are lacking the

654 local biological data to sufficiently evaluate past and future ecosystem changes, a high rate of

655 ocean acidification as observed in the Chukchi Sea (Qi et al., 2022b, a), associated with potential

656 temperature-induced shifts in the carbonate chemistry cycle (e.g. Orr et al. 2022), have the

657 potential to impact species and ecosystems. Exposure to low pH increases organismal energy

658 requirements for maintenance (e.g. acid-base regulation: Stumpp et al., 2012, compensatory

659 calcification: Ventura et al., 2016). Organisms can cope with increased energy costs using a

variety of strategies, ranging from individual physiological to behavioral responses, depending
on trophic level, mobility, and other ecological factors. For example, they can use available
stored energy to compensate for increased costs or they can decrease their metabolism to limit
costs (AMAP 2018). At the CEO, the low pH period observed during the summer and fall is
associated with elevated temperature and an elevated food supply for herbivores (Lalande et al.,
2020). The high availability of food may then foster compensation for the higher energetic costs
associated with exposure to low pH. However, a longer period of low pH as suggested by our
data could lead to a mismatch between the low pH and food availability, with cascading negative
consequences for the ecosystem (Kroeker et al., 2021). In winter, the low pH conditions are
associated with low temperature, no light, and low food level concentrations. These conditions
are likely to keep metabolisms low and limit the negative effects of exposure to low pH
(Gianguzza et al., 2014). As food availability is limited by the absence of light, this strategy may
be compromised by an increase in temperature that could also lead to increased metabolism.
Additional work is needed to understand impacts of acidification conditions and variability on
the marine biota of the Chukchi Sea, including field and laboratory experiments that evaluate
biological response under realistic scenarios. The characterization of the environmental
conditions at the CEO, including the variability in time, can be used to design single and multiple
stressor experiments (carbonate chemistry, temperature, salinity, food, oxygen; Boyd et al.

2018).

Indigenous communities are at the forefront of the changing Arctic, including changes in

accessibility, availability, and condition of traditional marine foods (Buschman and Sudlovenick,
2022; Hauser et al., 2021). Several marine species are critical to the food and cultural security of
coastal Inupiat who have thrived in Arctic Alaska for millenia. While it is not possible to resolve
the consequences of the seasonal and interannual variations in carbonate chemistry documented
in this manuscript without a proper sensitivity evaluation, the seasonally low pH conditions have
the potential to impact organisms like bivalves in a foraging hotspot for walrus (Jay et al., 2012;
Kuletz et al., 2015). Walrus, as well as their bivalve stomach contents, are important nutritional,
spiritual, and cultural components, raising concerns for food security in the context of ecosystem
shifts associated with the variability and multiplicity of climate impacts within the region (ICC,

2015).


**5. Concluding Thoughts**

The Chukchi Sea is undergoing a rapid environmental transformation with potentially
far-reaching consequences across the ecosystem. While we are lacking a long-term time-series,
we used this dataset to investigate the drivers of extreme pH, $\Omega_{arag}$, and $p\mathrm{CO}_2$ and document
conditions that could affect the ecological niches of organisms, including a fast rate of ocean
acidification, elongated sea ice free periods, increased primary productivity and elevated
temperature. While a combination of experimental and monitoring approaches is needed for an
understanding of the ecological consequences of these changes, our results also highlight the
urgency to mitigate $\mathrm{CO}_2$ emissions and simultaneously support Indigenous-led conservation
measures to safeguard an ecosystem in transition. Indigenous People in the Arctic have
established strategies to monitor, adapt to, and conserve the ecosystems upon which they depend.
Ethical and equitable engagement of Indigenous Knowledge and the communities at the forefront
of climate impacts can help guide research and conservation action by centering local priorities
and traditional practices, thereby supporting self-determination and sovereignty (Buschman and
Sudlovenick, 2022).

**Data availability**

The inorganic carbon data used in this manuscript are publicly available (Hauri and Irving, 2023a; Hauri and Irving, 2023b).


**Author contributions**

CH and BI managed and serviced the HydroC $CO_2$ and SeapHOx sensors, analyzed and published the data, and wrote the manuscript. SD and Peter Shipton carried out the CEO mooring deployments and recoveries and managed and serviced the CTD and $NO_3$ sensors. RP, DH, SD, and SLD contributed to the manuscript.

716

**Competing interests**

The authors have no competing interests.

719

**Acknowledgments**

The Chukchi Ecosystem Observatory is located on the traditional and contemporary hunting grounds of the Northern Alaska Iñupiat. We also acknowledge that our Fairbanks-based offices are located on the Native lands of the Lower Tanana Dena. The Indigenous Peoples of this land never surrendered lands or resources to Russia or the United States. We acknowledge this not only because we are grateful to the Indigenous communities who have been in deep connection with the land and water for time immemorial, but also in recognition of the historical and ongoing legacy of colonialism. We are committed to improving our scientific approaches and working towards co-production for a better future for everyone.

We acknowledge the World Climate Research Programme, which, through its Working
Group on Coupled Modelling, coordinated and promoted CMIP6. We thank the climate
modeling groups for producing and making available their model output, the Earth System Grid
Federation (ESGF) for archiving the data and providing access, and the multiple funding
agencies who support CMIP6 and ESGF.

**Financial support**
We would like to thank the National Pacific Research Board Long-term Monitoring
(NPRB LTM) program (project no. 1426 and L-36), the Alaska Ocean Observing System (award
no. NA11NOS0120020 and NA16NOS0120027), and the University of Alaska Fairbanks for
their financial support. Claudine Hauri, Brita Irving, and Seth Danielson also acknowledge
support from the National Science Foundation Office of Ocean Sciences and Polar Programs
(OCE-1841948 and OPP-1603116). Projects that assisted in the servicing of the CEO and/or
collected water column calibration data were funded by the National Science Foundation, Bureau
of Ocean Energy Management, National Oceanic and Atmospheric Administration, National
Oceanographic Partnership Program, and Shell Exploration and Production Company, Inc.
Maintenance and calibration of the CEO sensors is only possible due to the kind support of
numerous collaborators within the Arctic research community who helped with CEO deployment
and recovery or collected sensor calibration samples. We would therefore like to thank Peter
Shipton, Carin Ashjian, Jessica Cross, Miguel Goñi, Jackie Grebmeier, Burke Hales, Katrin Iken,
Laurie Juranek, Calvin Mordy, and Robert Pickart.

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

**Tables**
**Table 1.** Chukchi Ecosystem Observatory location and instrument sampling frequency. Sensor
type and parameter measured (italicized) shown in top row. Values in parenthesis indicate the
number of measurements averaged over the measurement interval window.

| Deployment | Latitude | Longitude | SUNA *NO₃* | HydroC CO2 *pCO₂* | SBE16 *CTD+* | SBE37 *CTD* | SeaFET *pH* | SBE63 *O₂* |
|---|---|---|---|---|---|---|---|---|
| 2016-2017 | 71.5996 | -161.5184 | 1 h | 12 h (300/5 min)* | 1 h | - | - | - |
| 2017-2018 | 71.5997 | -161.5189 | 1 h | 12 h (5/5 min) | 2 h | 2 h | 2 h (30/5 min) | 2 h |
| 2018-2019 | 71.5999 | -161.5281 | 1 h | 24 h (5/5 min) | 1 h | 2 h* | - | 2 h* |

| 2019-2020 | 71.5997 | -161.5275 | 1 h | 12 h (5/5 min) | 2 h | - | - | - |

* indicates the sensor did not return data over the whole year due to battery failure

CTD+ indicates ancillary data was available with the SBE16 file (e.g., chlorophyll fluorescence)




**Table 2.** Evaluation of $pH_{SeaFET}$ and $pH^{est}$ using reference pH from nearby discrete samples
($pH^{disc}_{calc}$). Uncertainty, $u_c$, is the propagated combined standard uncertainty from *errors.m* (Orr
et al., 2018). $pH_{SeaFET}$ and $pH^{est}$ were interpolated to the discrete timestamp. Figure S1 for
visualization of reference values.

| Date | Cruise | Cast No. | Distance (km) | $pH^{disc}_{calc} \pm u_c$ | Anomaly ($pH^{est}$-$pH^{disc}_{calc}$) | Anomaly ($pH_{SeaFET}$-$pH^{disc}_{calc}$) | Source |
|---|---|---|---|---|---|---|---|
| 2017-09-10 | HLY1702 | 127 | 0.52 | 8.0123±0.0166 | -0.0450* | -0.0354 | Cross et al., 2020a |
| 2019-08-11 | HLY1901 | 39 | 3.75 | 7.6423±0.012 | 0.0079* | - | Cross et al., 2021 |
| 2019-08-19 | OS1901 | 33 | 0.27 | 7.7367±0.0145 | -0.0200 | - | unpublished |

* indicates $pH^{disc}_{calc}$ was interpolated to mooring depth



**Figures**

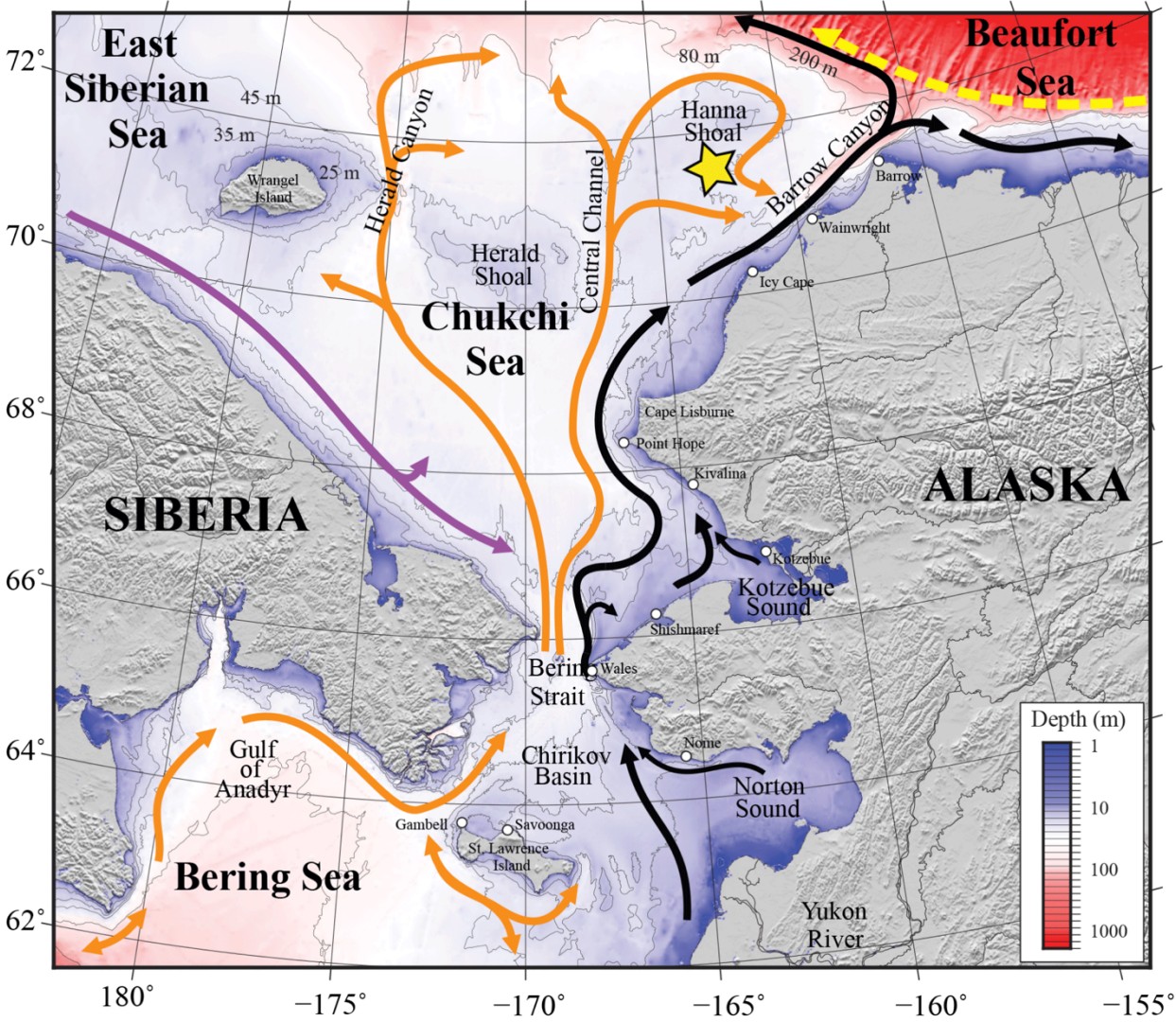


**Figure 1.  Map of the study area.** Bathymetry of the Chukchi, northern Bering, East Siberian
and eastern Beaufort seas is shown in color. The Chukchi Ecosystem Observatory (CEO)
location near Hanna Shoal is marked with a yellow star. General circulation patterns are shown
with arrows: black – Alaskan Coastal Water and Alaskan Coastal Current, dividing into the
Shelf-break Jet (right) and Chukchi Slope Current (left, Corlett and Pickart, (2017)); orange –
Anadyr, Bering, and Chukchi Seawater; purple – Siberian Coastal Current; yellow – Beaufort
Gyre boundary current. Figure is from Hauri et al. (2018).


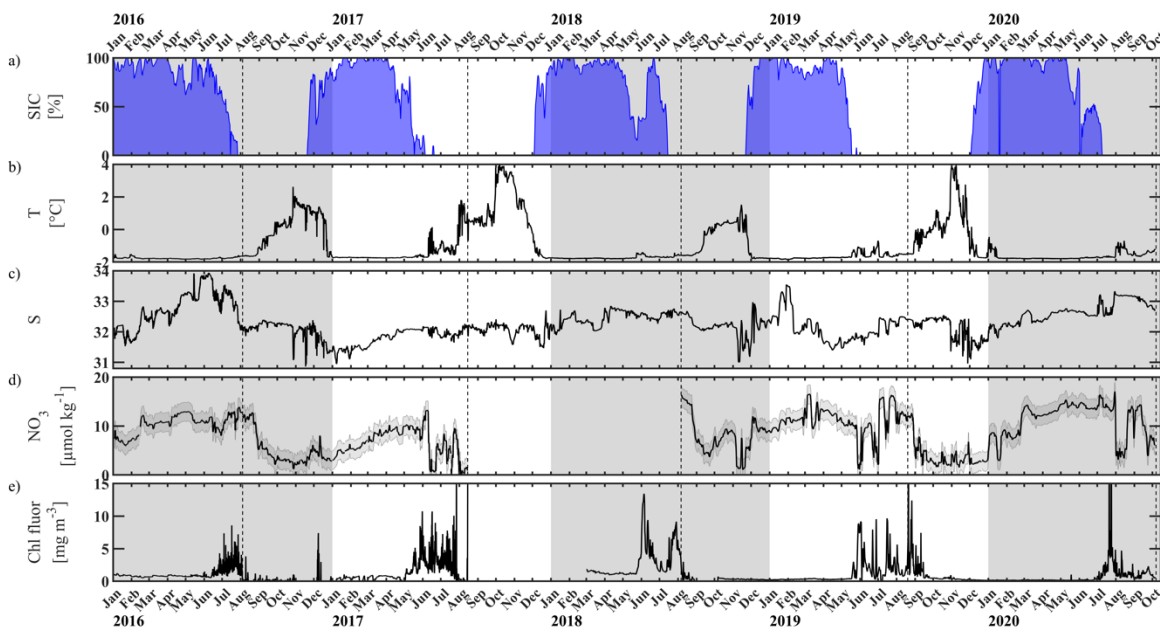


**Figure 2. Chukchi Ecosystem Observatory timeseries from 2016 through 2020.** a) sea ice
concentration (blue shading to highlight coverage, %; DiGirolamo et al., 2022), b) temperature
(°C), c) salinity, d) NO₃ with uncertainty envelope (μmol kg⁻¹), and e) chlorophyll fluorescence
(mg m⁻³). Years are indicated by alternating gray and white background shading. The vertical
dotted gray lines indicate the mooring turn around timing.

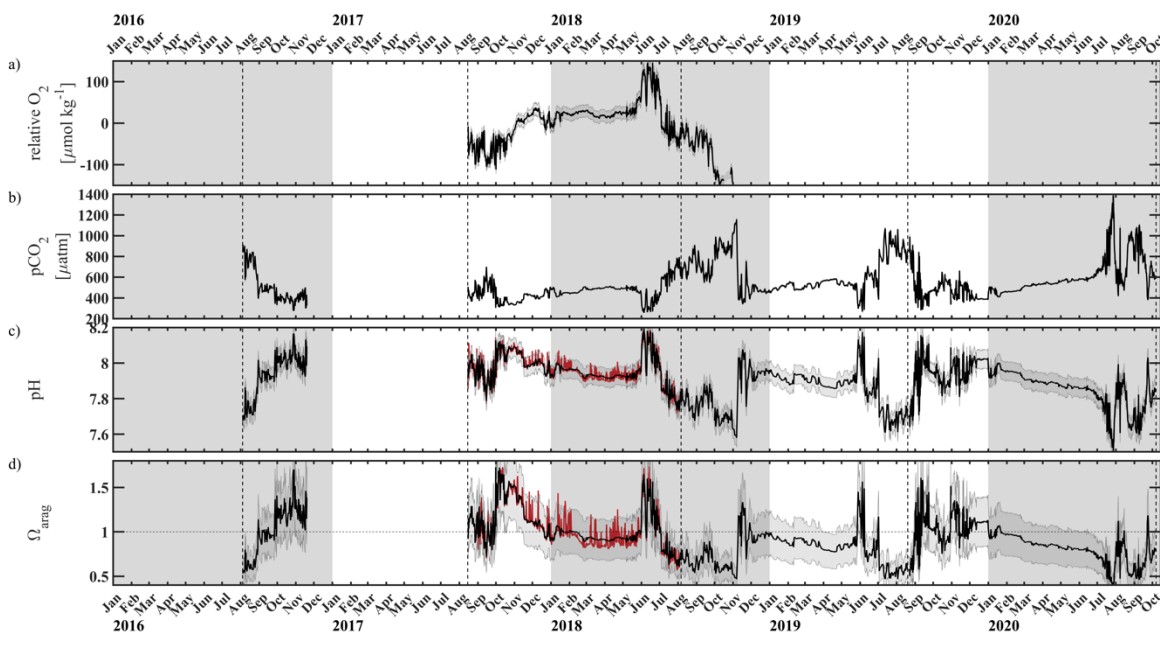


**Figure 3. Chukchi Ecosystem Observatory timeseries from 2016 through 2020, part 2.** a)
relative dissolved oxygen with uncertainty envelope (relative to the mean; μmol kg$^{-1}$), b) $p$CO$_2$
with uncertainty envelope (μatm; Hauri and Irving, 2023a), c) pH with uncertainty envelope
(pH$^{est}$ in black, pH$_{SeaFET}$ in red; Hauri and Irving 2023b), and d) aragonite saturation state with
uncertainty envelope ($\Omega_{arag}$($p$CO$_2$, pH$^{est}$) in black; $\Omega_{arag}$($p$CO$_2$, pH$_{SeaFET}$) in red). Years are
indicated by alternating gray and white backgrounds. The vertical dotted gray lines indicate the
mooring turn around timing.

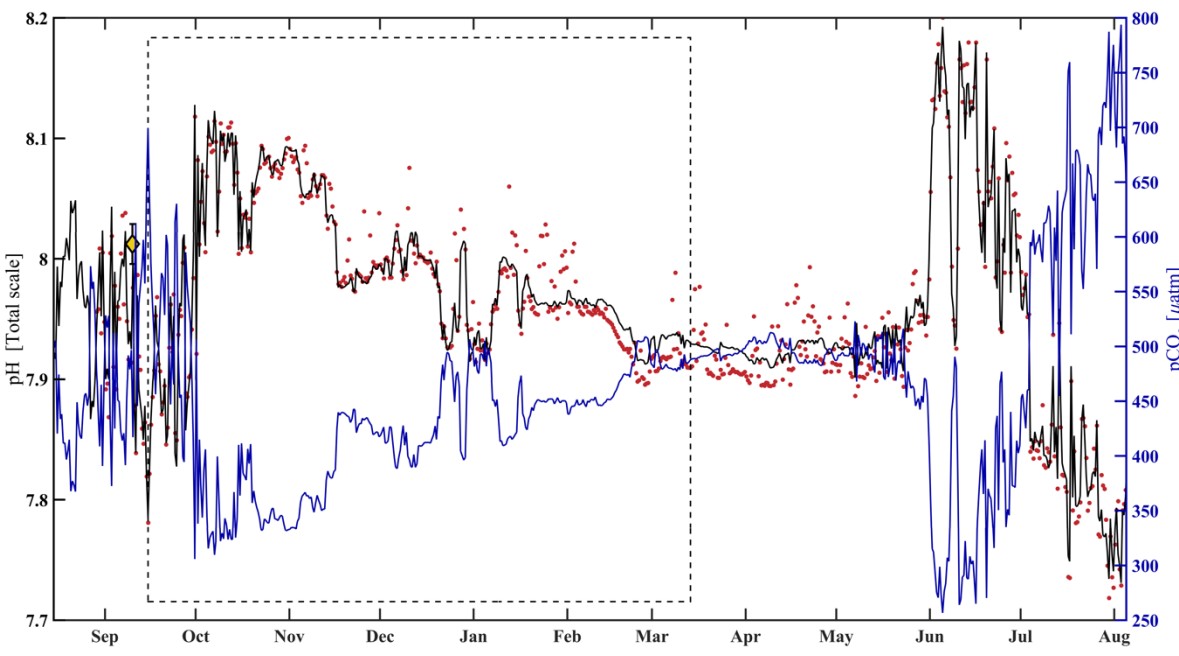



**Figure 4. HydroC $pCO_2$ and pH highlighting mirrored trend from mid-August 2017 to**
**beginning of August 2018.** Measured pH (pH$_{SeaFET}$, red dots) is interpolated onto the HydroC
$pCO_2$ timestamp (blue), and pH$^{est}$ is shown as the solid black line. The dashed box shows the
period over which pH$^{est}$ was trained. The yellow faced diamond with error bars show reference
pH$^{disc}_{calc}$ ± u$_c$ (Table 2; Cross et al., 2020a; Orr et al., 2018).


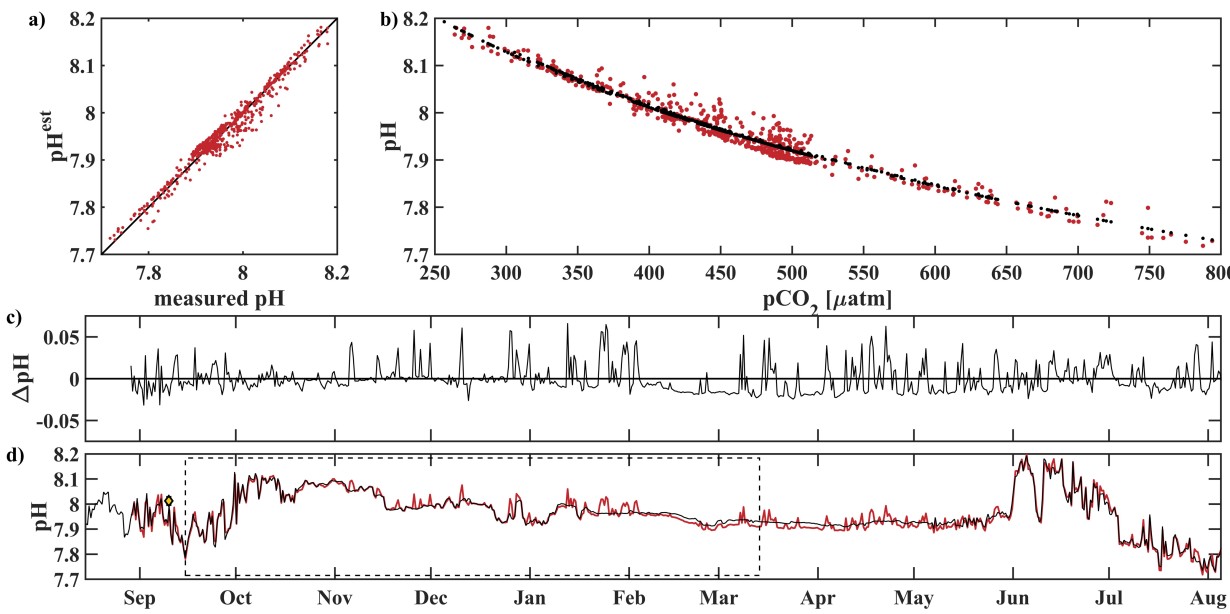


**Figure 5**. **Performance of the pH algorithm.** (a) $pH_{SeaFET}$ vs $pH^{est}$ with black line highlighting
1:1 ratio, (b) $pCO_2$ vs $pH_{SeaFET}$ (red) and $pCO_2$ vs $pH^{est}$ (black), (c) residual pH ($pH_{SeaFET}$ −
$pH^{est}$), and (d) $pH_{SeaFET}$ (red) and $pH^{est}$ (black) vs. time, with dashed box highlighting the period
over which $pH^{est}$ was trained (15 September - 14 March 2017), and the yellow faced diamond
with error bars showing reference $pH^{disc}_{calc} \pm u_c$ (Table 2; Cross et al., 2020).




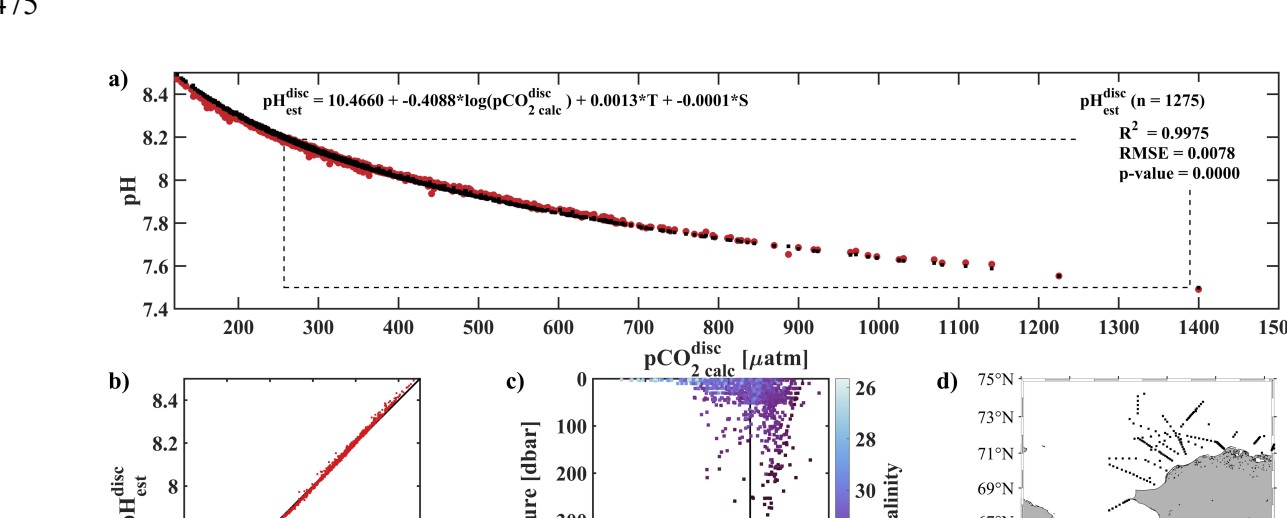


**Figure 6. Evaluation of the pH algorithm.** pH$^{est}$ evaluation with pH$^{disc}_{calc}$ from discrete
samples collected during 4 cruises in the fall or early winter (August - November) of 2017-2020
and pH$^{disc}_{est}$ from our linear regression model (Equation 2). (a) $p$CO$_2$$^{disc}_{calc}$(TA, DIC) vs pH (red
pH$^{disc}_{calc}$ and black pH$^{disc}_{est}$) with dashed black box showing the range of pH and $p$CO$_2$ observed
at the CEO at 33 m depth, (b) pH$^{disc}_{calc}$ vs pH$^{disc}_{est}$ with black 1:1 ratio, (c) residual pH (pH$^{disc}_{calc}$ -
pH$^{disc}_{est}$) vs depth with color shading by salinity and black vertical line at 0, and (d) map showing
the locations of the 1275 discrete water samples used for evaluation (Monacci et al., 2022; Cross
et al., 2021; 2020a; 2020b).


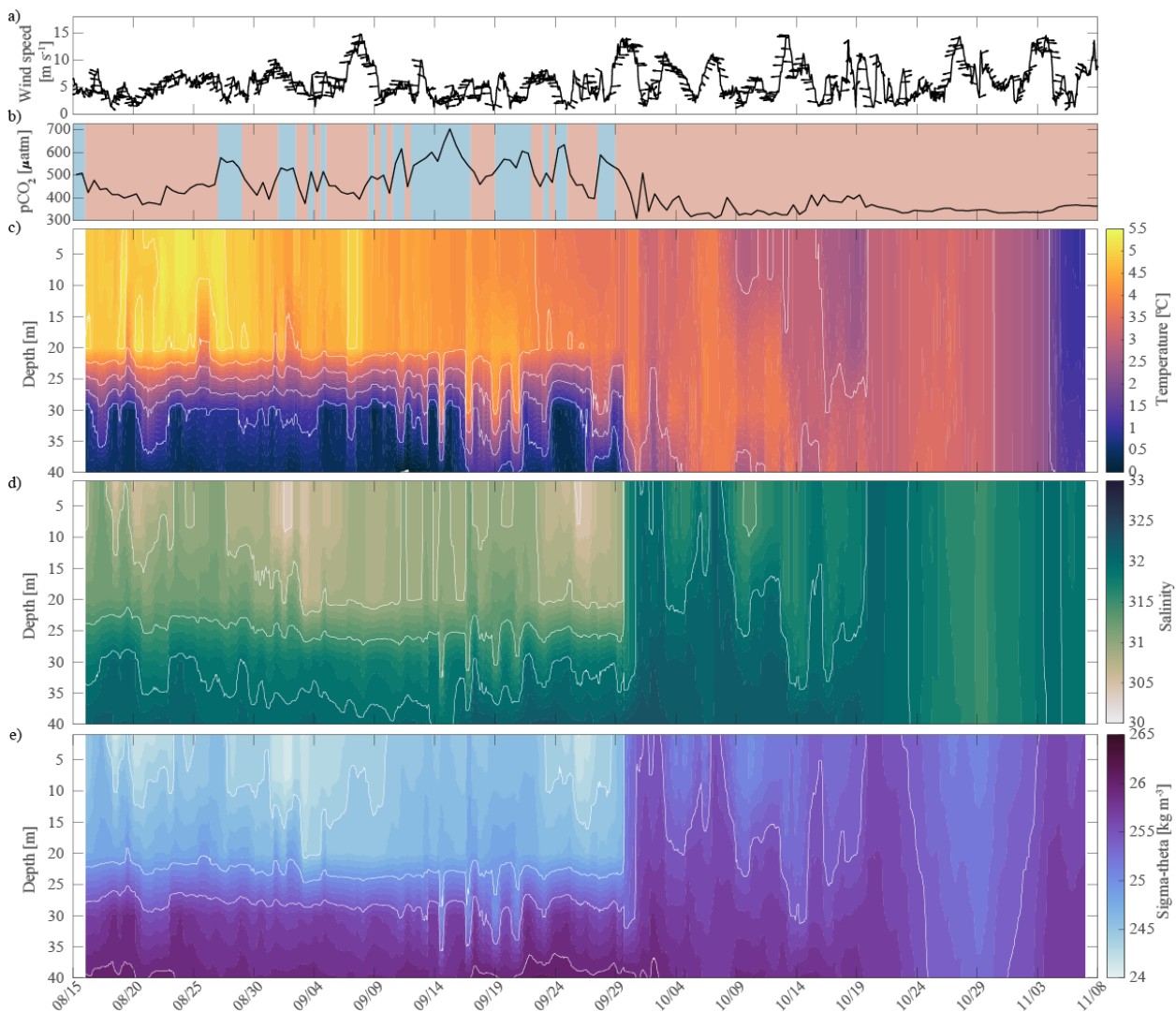


**Figure 7. Water column structure from late summer 2017 to freeze up.** Profiles of a) wind
speed and direction (arrows pointing downwind) from the NOAA-operated Wiley Post-Will
Rogers Memorial Airport, b) $p\mathrm{CO}_2$ (μatm) with blue background indicating the water was
undersaturated regarding aragonite ($\Omega_\mathrm{arag} < 1$) and red shading indicating aragonite
oversaturation ($\Omega_\mathrm{arag} >= 1$), c) temperature (°C), d) salinity, and e) sigma-theta (kg m$^{-3}$).
Temperature (c) and salinity (d) were measured at 8, 20, 30, and 40 m by the Chukchi Ecosystem
Observatory freeze-up detection mooring deployed in fall 2017. Density was calculated with the
TEOS-10 GSW Oceanographic Toolbox (McDougall and Baker, 2011).

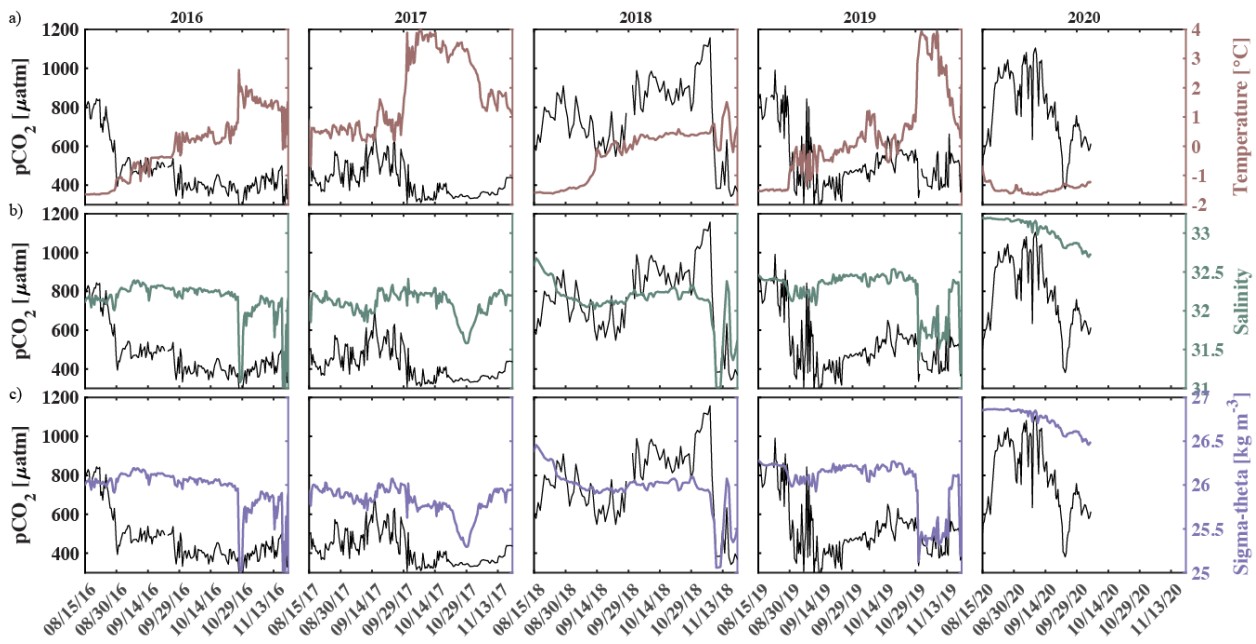


**Figure 8. Impact of water column mixing on *p*CO₂.** Timeseries of *p*CO₂ (black, left axis) and
a) temperature (maroon, right axis), b) salinity (green, right axis), and c) density (purple, right
axis) for 15 August to 1 December in 2016 -2020 measured at ~33m septh at the Chukchi Sea
Ecosystem Observatory.







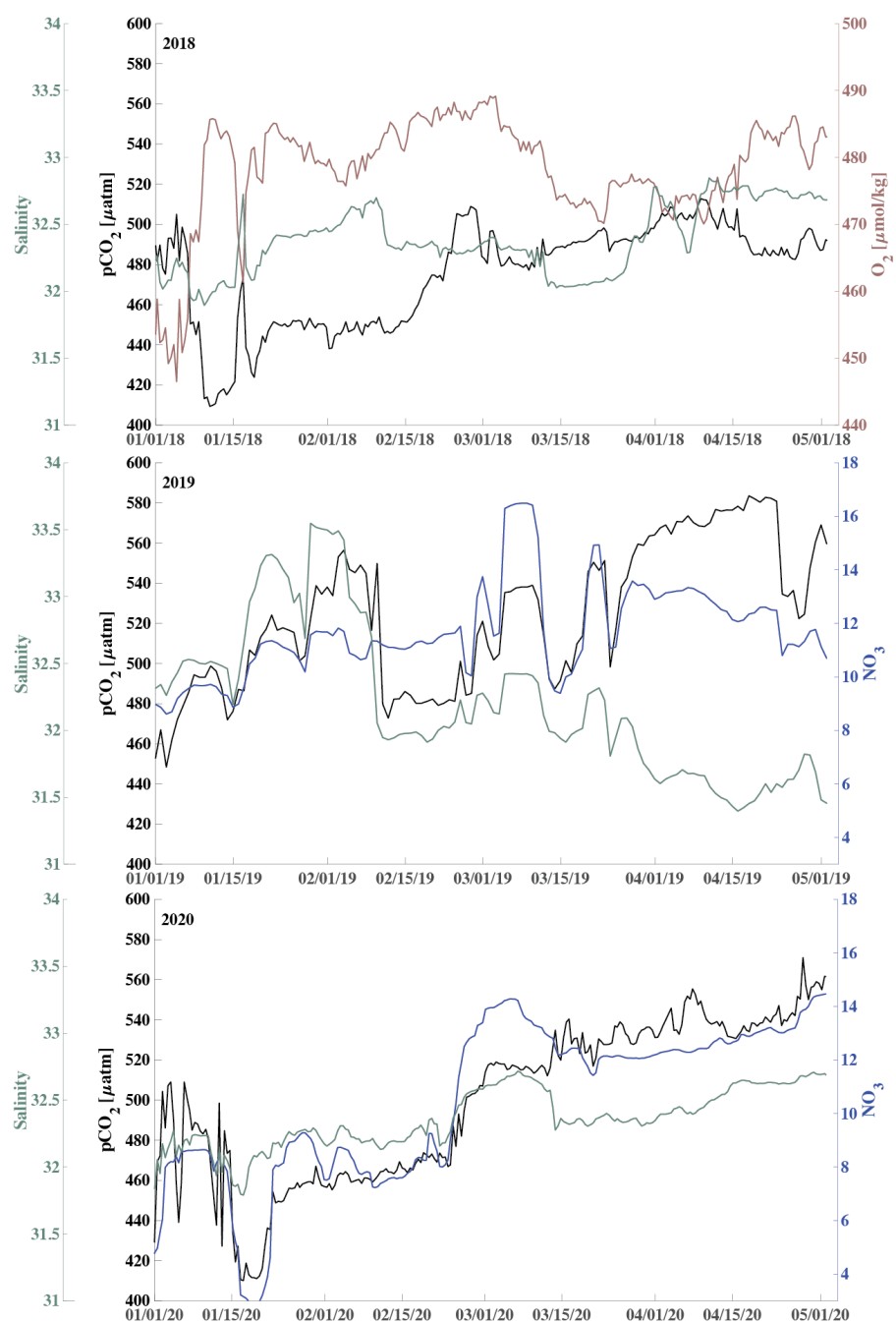


**Figure 9. Respiration under the sea ice.** Timeseries of $p$CO$_2$ (black) and salinity (green, left

axis), and oxygen (O$_2$, µmol kg$^{-1}$, maroon, top) and nitrate (NO$_3$, µmol kg$^{-1}$, blue, middle and

bottom) concentration (right axis during January through April for 2018 (top), 2019 (middle) and

2020 (bottom).


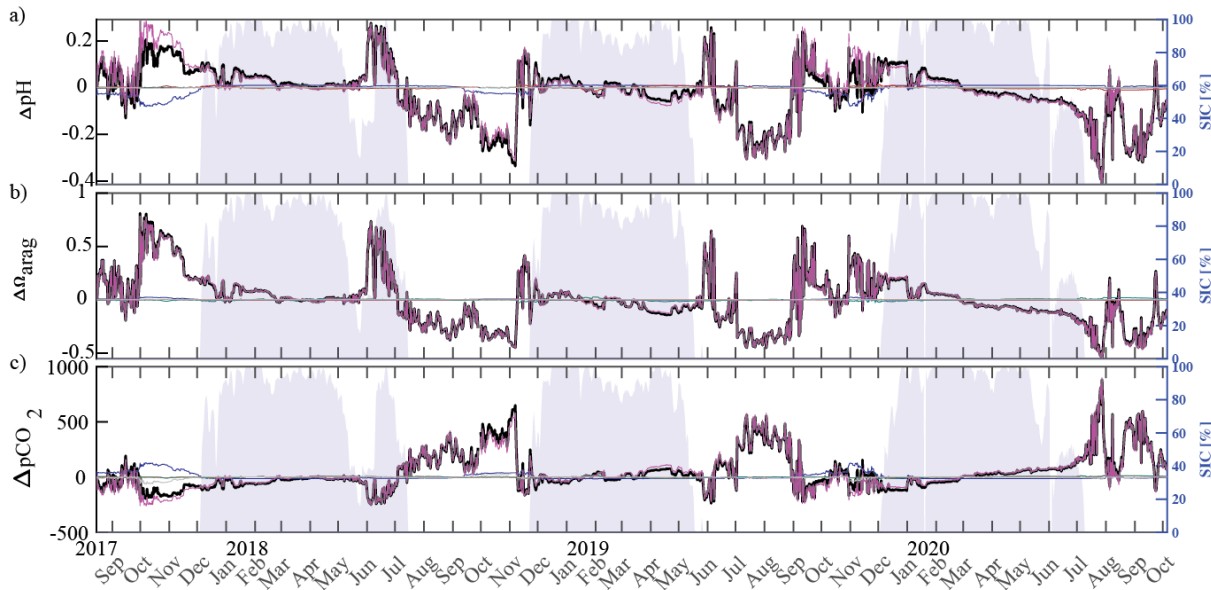


**Figure 10. Drivers of the inorganic carbon system.** Component timeseries of the linear Taylor
decomposition of a) pH, b) $\Omega_{arag}$, and c) $pCO_2$. The perturbation effects due to salinity (red),
temperature (blue), biogeochemistry (pink), and freshwater mixing (green), and an estimated
residual term (grey) were computed following Rheuban et al. (2019). Sea ice concentration (blue
shading, %; DiGirolamo et al., 2022) is shown on the right axes.

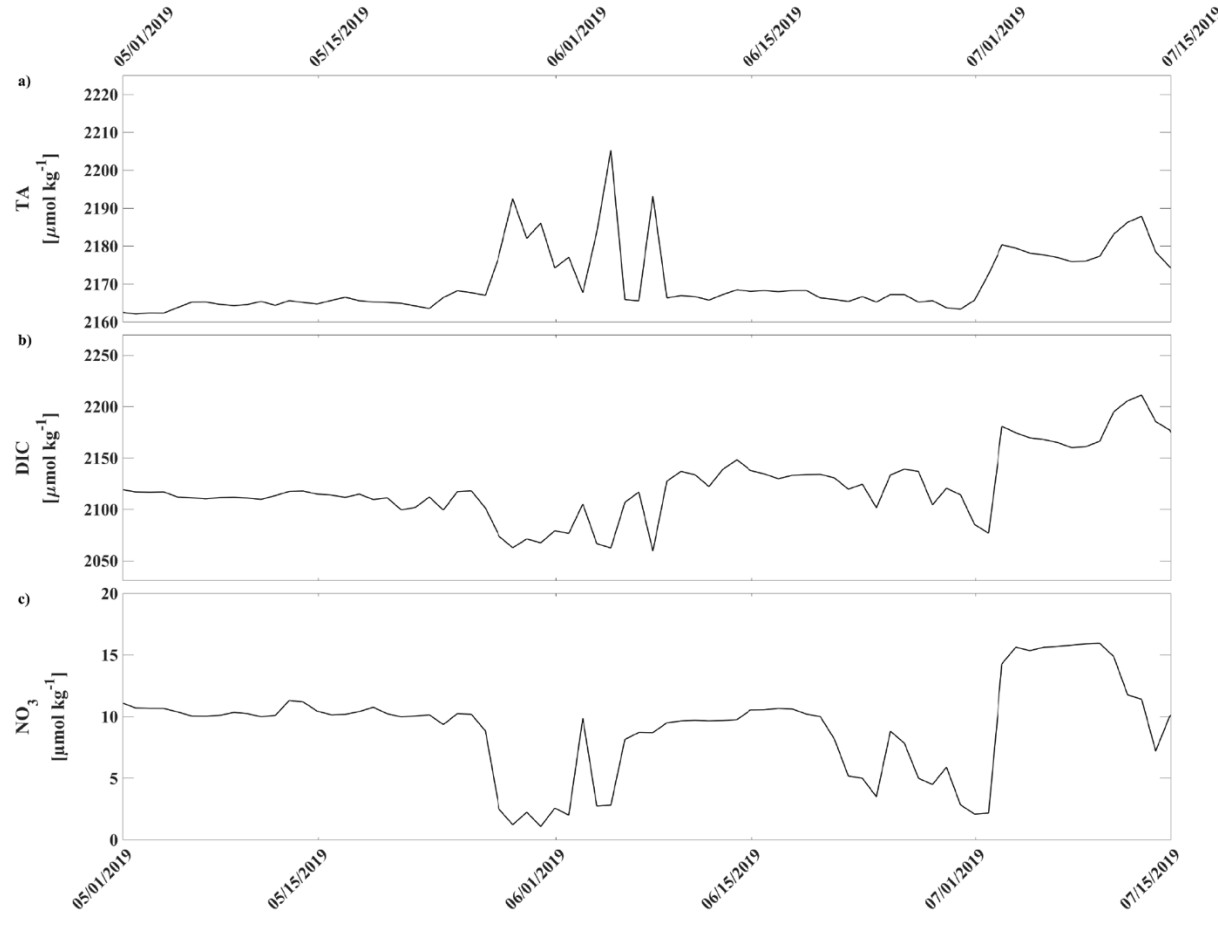


**Figure 11. Spring 2019 relaxation event.** Timeseries of a) total alkalinity (TA, μmol kg⁻¹), b)
dissolved inorganic carbon (DIC, μmol kg⁻¹), and c) nitrate (NO₃, μmol kg⁻¹) from May 1st, 2019
through July 15th, 2019.

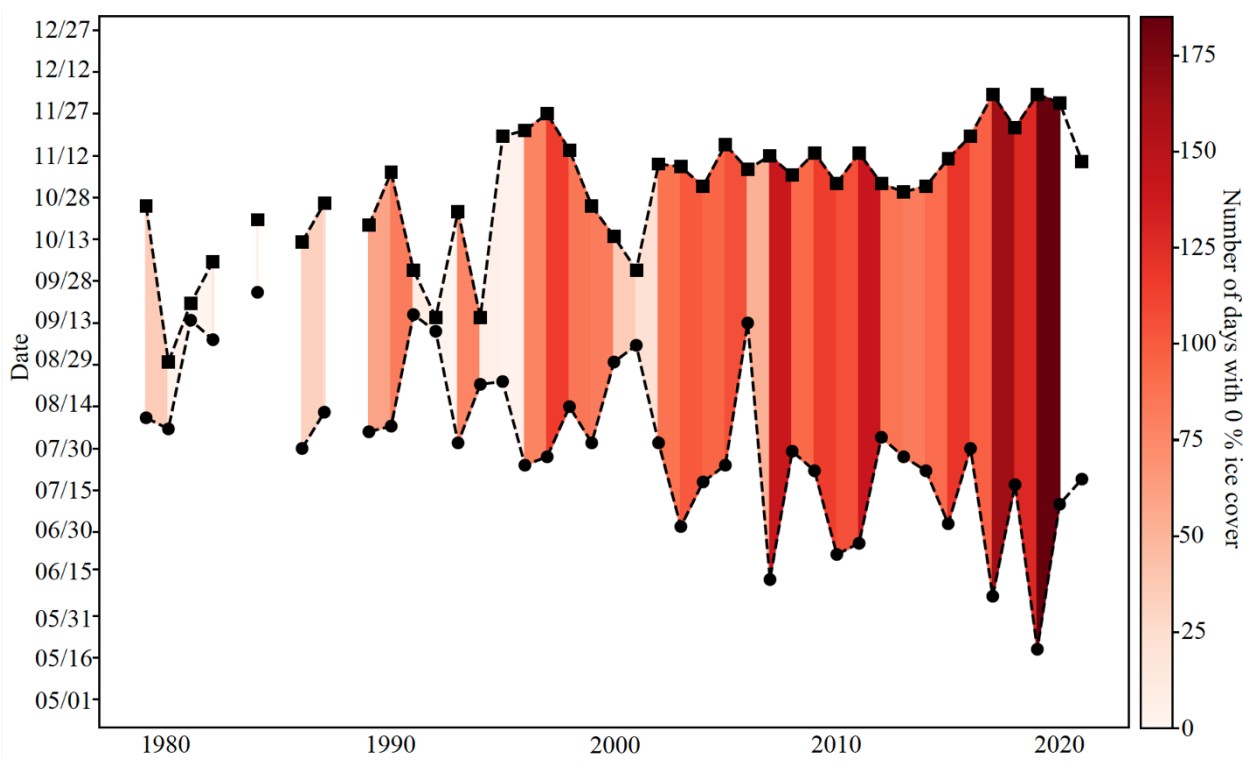



**Figure 12. Low sea ice period at the Chukchi Sea Observatory.** Timeseries of start (circle)
and end (square) of low sea ice (< 15 % per grid cell) period from 1982-2021. Shades of red
illustrate number of days with 0 % sea ice cover. The satellite sea ice cover at the observatory
site was taken from the NSIDC (DiGirolamo et al., 2022).