# Peer review of "Insights into carbonate environmental conditions in the Chukchi Sea"

_EGUsphere, 2023_

## Referee Comment (RC1)

Review of the manuscript "Sea ice loss translates into major shifts in the carbonate environmental conditions in Arctic Shelf Sea" by Hauri et al., 2023

**Summary:**
The manuscript reported time-series observations of a suite of hydrographic and biogeochemical parameters (T, S, $p$CO$_2$, pH, NO$_3$) in the subsurface water (at 33 m depth) in the Chukchi Sea from 2016 to 2020. The dataset is very interesting and valuable for better understanding the rapid changes in carbonate chemistry, ocean acidification, as well as the impacts on Arctic organisms and the ecosystem. The authors focused on examining seasonal and interannual variability and discussing the controls of the carbonate system. They found that the subsurface waters have very high CO$_2$, low pH and $\Omega$ in summer and fall, which is due to remineralization of organic matter. They further identified two periods (spring and late fall) with relatively lower CO$_2$ and higher pH and $\Omega$, called "ocean acidification relaxation events". The time-series data well explained the causes and drivers for these events, which provides new insights into seasonal variations in the carbonate system in the Chukchi subsurface waters.

However, I cannot recommend the manuscript for publication in its current version because I have several concerns that need to be addressed. I listed my comments below:

**Major comments:**
**Title:**
I find that the title of this manuscript is a little bit too ambitious. One mooring location may not represent all Arctic Shelf Seas. Maybe consider changing "Arctic Shelf Sea" to "the Chukchi Sea".

**Salinity-normalized data**
Although the authors pointed out that freshwater can affect the inorganic carbon system and they attempt to normalize DIC and TA to a reference salinity, they didn't clarify the freshwater end-member they used for salinity normalization. Since normalized data are critical for further quantitative analysis, they should be clearly presented.

In addition, it is not clear to me if the authors used normalized data in Results and Discussion sections because they presented changes in DIC, TA, and NO$_3$, not nDIC, nTA, nNO$_3$, thus it is really hard to evaluate their data interpretation.

**Minor comments:**
Line 25 Need to define "$p$CO$_2$" the first time it appears.

Line 37 I feel that "CO$_2$-depleted surface water" is not a very accurate description. The authors need to explain how they define "depleted". Maybe "low $p$CO$_2$ surface water" is good enough.

Line 37 I feel that ice formation-induced water convection in late fall may also play a role.

Line 56 The citation is not complete.

Line 67-71 Please carefully check throughout the entire text and modify the terms as "$CaCO_3$", "$CO_2$", $CO_3^{2-}$, "$pCO_2$", and "$\Omega_{arag}$". I cannot list all the wrong spell here.

Line 95 Need to define CEO at the first time it appears

Line 190-192 Should Move to 2.5 CTD and Oxygen section. How about DO data collected from Aug 2017 to Aug 2018?

Line 220 What is CEO-2? Need to explain.

Line 220 …mooring near the HydroC…

Line 264 I don't understand why the $r^2$ for the validation part is even higher than that of the training part. Why select data from 15/9/2017 to 14/3/2018 as the training dataset?

Line 253-255 The parameter symbol of $\alpha$ (not a) should be consistent throughout the text and equation.

Line 288-289 TA, DIC, and $\Omega_{arag}$ (Figure 2 i-k) were calculated based on data from the HydroC $pCO_2$, pHest and S, T, and pressure (P) from the SBE16.

Line 306 (Sulpis et al. (2020) found that…

Line 323 Please cite the literature for supporting this statement "also at times by the Mackenzie River outflow from the eastern Beaufort Sea and the large Russian Arctic rivers located to the west of the Chukchi Sea."

Line 331 $nP = (P - P(s=0))/S * Sref + P(s=0)$,          (2)

Line 336 Not clear how the authors determine the freshwater endmember for DIC, TA, $pCO_2$, and $NO_3$. I suggest listing all estimated freshwater end-member in a table in the supporting information.

Line 337 I don't think $pCO_2$ can be directly normalized by salinity because it doesn't change linearly with salinity.

Line 336-338 How does the brine rejection process during ice formation affect seasonal variations in these variables? Especially during the winter.

Line 338 and 345 Should be (Figure S3) and (Figure S4)

Line 344 $pCO_{2,NT} = pCO_2 * \exp(0.0423(T_{ref} - T))$          (3)
Should list the equations in a number order

Line 346 …surface waters were entrained to…

Line 348 "$pCO_2$"

Line 357 Please explain why selected 51% of sea ice concentration as a threshold. People normally use <15% as a threshold for open water.

Line 375 Needs to define "springtime" in this study.
"higher pH and $\Omega_{arag}$ and lower $pCO_2$" compared to what? The overall means?

Line 377 I don't see the spikes in June of 2020

Line 384 If the oxygen data is already known as bad data as shown in Fig. S2, how can the author trust the oxygen changes can be used for quantitative analysis?

Line 388 Not clear how the authors estimated TA increase with the assumption of NO3 consumption. No observation of NO3 in the 2018 spring. Need more explanation.

Line 404 If the water column is well-stratified, does the observation at 33 m reflect the surface mixed layer or bottom layer at this period? If it is reflecting the variation of the surface mixed layer, the CO2 air-sea exchange should be considered. If it is reflecting the variations of bottom layer, how CaCO3 mineral dissolution in the surface layer increases TA in the bottom layer?

Line 407 Needs to define "fall" in this study
…drops in $pCO_2$ ?

Line 431 There is no figure (Figure 7) showing changes in $\Omega_{arag}$

Line 458 DIC changes or nDIC change?

Line 491 NO3 decreased below 10 $\mu$mol kg$^{-1}$ shown in Fig. 2f

Line 514 the authors need to clarify that they are comparing the acidification rate at the surface vs. in the water columns

Line 515-516 I didn't see the result of the statement "The weaker trend was calculated with data starting in 1994, whereas the stronger trend used data starting eight years later." in Qi et al. (2022). Need to explain how the author concluded that.

Line 520 Please clarify if the model results were derived from the depth of the CEO.

Line 604 …pH (e.g. Gianguzza et al., (2014).

Line 625… and lower $pCO_2$, were driven by a combination

Line 627-629 It is not necessarily and exclusively to be sea ice algae, because the water column is not well-stratified in the spring, the phytoplankton is also able to grow at the subsurface or to be entrained into the subsurface.
Please check Arrigo et al., 2017 and Ouyang et al., 2022

Arrigo, K. R., Mills, M. M., van Dijken, G. L., Lowry, K. E., Pickart, R. S., & Schlitzer, R. (2017). Late spring nitrate distributions beneath the ice-covered northeastern Chukchi Shelf. Journal of Geophysical Research: Biogeosciences, 122(9), 2409-2417.

Ouyang, Z., Collins, A., Li, Y., Qi, D., Arrigo, K. R., Zhuang, Y., ... & Cai, W. J. (2022). Seasonal Water Mass Evolution and Non-Redfield Dynamics Enhance $CO_2$ Uptake in the Chukchi Sea. Journal of Geophysical Research: Oceans, 127(8), e2021JC018326.

Line 693 "dataset"

Line1273 I suggest enlarging Figure 2 in the y-axis scale. Since this is the most important and valuable figure in the manuscript, it should be made more readable.
Also, it is really hard to see the measured pH (grey line) in Fig. 2c. Suggest changing it to another color.

Line 1286-1287 Please enlarge the green diamond and its error bars in the figure to make it more readable.

Supplement

Line 7-8 the format of the date should be consistent with the label in the figure.

Figure S3 Please enlarge the labels to make them more readable. Also, suggest changing the normalized data to a more readable color.

Figure S4 Suggests changing the temperature-normalized data to a more readable color.

---

## Author Response (AR1)

RC1

We would like to thank reviewer 1 for their constructive comments and appreciate their time to carefully read the manuscript. We are excited that they found the data and findings interesting and helpful to better understand the changing Chukchi Sea seascape. As a result of their review and others, we realize that the paper will require major revisions before it is ready for publication. Nevertheless, we believe that once it is revised it will make an important contribution to improve our knowledge about inorganic carbon dynamics in the Chukchi Sea and will fit well into the journal Biogeosciences. Our responses are shown in **bold** below.

**Major comments:**

Title:

I find that the title of this manuscript is a little bit too ambitious. One mooring location may not represent all Arctic Shelf Seas. Maybe consider changing "Arctic Shelf Sea" to "the Chukchi Sea".

**We changed the title to "Insights into carbonate environmental conditions in the Chukchi Sea"**

Salinity-normalized data

Although the authors pointed out that freshwater can affect the inorganic carbon system and they attempt to normalize DIC and TA to a reference salinity, they didn't clarify the freshwater end-member they used for salinity normalization. Since normalized data are critical for further quantitative analysis, they should be clearly presented.

In addition, it is not clear to me if the authors used normalized data in Results and Discussion sections because they presented changes in DIC, TA, and NO3, not nDIC, nTA, nNO3, thus it is really hard to evaluate their data interpretation.

**We rewrote the results and discussion sections. Results contain a description of the timeseries, whereas in the discussion we used a Taylor decomposition to analyze the potential impact of freshwater (Section 4.3).**

Minor comments:

Line 25 Need to define "$p\mathrm{CO}_2$" the first time it appears.

**We now define it in the abstract (L24) and then again in the text (L58).**

Line 37 I feel that "CO2-depleted surface water" is not a very accurate description. The authors need to explain how they define "depleted". Maybe "low pCO2 surface water" is good enough.

**Agreed. We reworded to "have a low $p$CO₂" here and throughout the manuscript.**

Line 37 I feel that ice formation-induced water convection in late fall may also play a role.

**Convection does drive some turnover, but the water column generally homogenizes due to wind mixing prior to the water column cooling to the freezing point, so pCO₂ is already low by the time ice forms at this site.**

Line 56 The citation is not complete.

**We redid the entire bibliography.**

Line 67-71 Please carefully check throughout the entire text and modify the terms as "CaCO3", "CO2", CO32−, "$p$CO2", and "Ωarag". I cannot list all the wrong spell here.

**Done throughout manuscript.**

Line 95 Need to define CEO at the first time it appears

**We now introduce it on line 94: "The Chukchi Ecosystem Observatory (CEO) is situated in a benthic hotspot where high primary production supports rich and interconnected benthic and pelagic food webs (Grebmeier et al., 2015; Moore and Stabeno, 2015)."**

**2.1 And in the subtitle on line 147: "The Chukchi Ecosystem Observatory (CEO)"**
Line 190-192 Should Move to 2.5 CTD and Oxygen section. How about DO data collected from Aug 2017 to Aug 2018?

**We rewrote section 2.5:**

> **Two CTDs were deployed on the CEO mooring near the HydroC CO₂ depth. The main pumped Sea-Bird SeaCAT (SBE16) has been deployed on the CEO mooring around 33 m depth since 2014. A pumped SBE43 oxygen sensor was deployed with the SBE16 during the 2015-2016, 2017-2018, and 2019-2020 deployments but only data returns from the 2017-2018 deployment is discussed briefly in this manuscript (Figure S2).**
> **The other pumped CTD was a Sea-Bird MicroCAT (SBE37-SMP-ODO), which was integrated with an optical dissolved oxygen sensor (SBE63; Figure S2), and the SeaFET pH sensor within the SeapHOx instrument. The SeapHOx was deployed in fall 2016, 2017, and 2018. The SBE37-SMP-ODO did not record any CTD or oxygen data during the 2016 deployment and only recorded CTD and oxygen data between August and November 3 in 2018 due to battery failure.**

Processing of these data included temperature and conductivity correction using pre- and post-calibration data following Sea-Bird Application Note 31 and oxygen correction using pre- and post-calibration data following Sea-Bird Module 28. Oxygen was converted from ml/l to µmol/kg following Bittig et al. (2018). Density and practical salinity were calculated using the TEOS-10 GSW Oceanographic Toolbox (McDougall and Baker, 2011).

Differences between the two oxygen sensors (SBE43 and SBE63) of approximately 145 to 265 µmol/kg were observed over the 2017-2018 deployment, and both moored sensors had varying offsets compared to nearby casts (Figure S2).  Therefore, only relative oxygen values from the freshly calibrated SBE63 are discussed in this paper.

The freeze-up detection mooring (Figure 6) consisted of four Sea-Bird SBE 37 inductive modem CTD sensors that transmitted in real time hourly temperature, salinity, and pressure data via the surface float from four subsurface depths (8, 20, 30, and 40 m; Hauri et al., 2018)."

Line 220 What is CEO-2? Need to explain.

**We deleted this because it is not relevant for the reader.**

Line 220 …mooring near the HydroC…

**Done.**

Line 264 I don't understand why the r2 for the validation part is even higher than that of the training part. Why select data from 15/9/2017 to 14/3/2018 as the training dataset?

**The training length of 180 days was chosen to cover approximately half of the measured pH timeseries, and to allow cross-validation of the algorithm using the remaining data outside the training period. The specific date range of 15/9/2017-14/3/2018 was arbitrarily selected as the training range so as not to bias the algorithm performance, except that it was intentionally shifted to start closer to the start of the measured pH timeseries so that biofouling (very little observed at the CEO) or sensor drift influence was minimized.**

**We revised the text to make this clearer L277: "…the algorithm was trained over a arbitrarily 180-day time period (15/9/2017-14/3/2018, Figure 3, shaded area)."**

**Though not directly addressed in the paper, we believe that the different drivers in the inorganic carbon system throughout the year play a role in why the R2 is different over the training period (sea ice free mid-summer through the stormy season into mid-winter) vs the cross-validation period (mid- winter through sea-ice breakup and into mid-summer).**

Line 253-255 The parameter symbol of $\alpha$ (not a) should be consistent throughout the text and equation.

**Done.**

Line 288-289 TA, DIC, and Ωarag (Figure 11 a & b and Figure 3d) were calculated based on data from the HydroC pCO2, pHest and S, T, and pressure (P) from the SBE16.

**We rephrased to (L287):** "**TA, DIC, and $\Omega_{arag}$ (Figure 11 a & b and Figure 3d) were calculated based on measured $p$CO$_2$, S, T, and pressure (P) and algorithm-based pH (pH$^{est}$).**

Line 306 (Sulpis et al. (2020) found that…

**Done.**

Line 323 Please cite the literature for supporting this statement "also at times by the Mackenzie River outflow from the eastern Beaufort Sea and the large Russian Arctic rivers located to the west of the Chukchi Sea."

**We added the following citations:**

**Asahara, Y., Takeuchi, F., Nagashima, K., Harada, N., Yamamoto, K., Oguri, K., and Tadai, O.: Provenance of terrigenous detritus of the surface sediments in the Bering and Chukchi Seas as derived from Sr and Nd isotopes: Implications for recent climate change in the Arctic regions, Deep Sea Res. Part II Top. Stud. Oceanogr., 61–64, 155–171, https://doi.org/10.1016/j.dsr2.2011.12.004, 2012.**

**Jung, J., Son, J. E., Lee, Y. K., Cho, K.-H., Lee, Y., Yang, E. J., Kang, S.-H., and Hur, J.: Tracing riverine dissolved organic carbon and its transport to the halocline layer in the Chukchi Sea (western Arctic Ocean) using humic-like fluorescence fingerprinting, Sci. Total Environ., 772, 145542, https://doi.org/10.1016/j.scitotenv.2021.145542, 2021.**

Line 331 nP = (P -P(s=0))/S* Sref +P(s=0), (2)

**This section has been removed.**

Line 336 Not clear how the authors determine the freshwater endmember for DIC, TA, $p$CO2, and NO3. I suggest listing all estimated freshwater end-member in a table in the supporting information.

**The freshwater endmembers are now discussed in section 4.3.**

Line 337 I don't think pCO2 can be directly normalized by salinity because it doesn't change linearly with salinity.

**Thank you for pointing this out. We redid the analysis with a Taylor decomposition following Rheuban et al., 2019 (section 4.3).**

Line 336-338 How does the brine rejection process during ice formation affect seasonal variations in these variables? Especially during the winter.

**We did not find any signs of brine rejection.**

Line 338 and 345 Should be (Figure S3) and (Figure S4)

**Figures S3 and S4 are no longer needed.**

Line 344 pCO2 ,NT = $p$CO2 * exp(0.0423(Tref - T)) (3)

Should list the equations in a number order

**Was revised accordingly.**

Line 346 …surface waters were entrained to…

**Was revised accordingly.**

Line 348 "pCO2"

**Was revised accordingly.**

Line 357 Please explain why selected 51% of sea ice concentration as a threshold. People normally use <15% as a threshold for open water.

**We adjusted the figure using <15 % as a threshold. It does not affect the broader results.**

Line 375 Needs to define "springtime" in this study.

**The different seasons used in the paper are now defined (L381): "In the following we will focus on spikes of high pH and $\Omega_{arag}$ and low $p$CO$_2$ that occur in spring (May-June) and fall (September-December); we define these spikes as relaxation events (see discussion for justification of term)."**

"higher pH and Ωarag and lower $p$CO2" compared to what? The overall means?

**It now reads (L384):" Springtime relaxation events at 33 m depth that exhibit relatively higher pH and $\Omega_{arag}$ and lower $p$CO$_2$ compared to the overall mean, are likely consequences of the dissolution of CaCO$_3$ minerals and photosynthetic activity during sea ice break-up (Figure 2)."**

Line 377 I don't see the spikes in June of 2020

**We address this in section 3.3 (L456):**

**"3.4 Spring and summer of 2020 were different**

The seasonal cycle in 2020 strongly contrasted with the previous observed years. $pCO_2$ gradually increased by roughly 200 µatm throughout the sea ice covered months to 650 µatm when sea ice started to retreat at the beginning of July. By the end of July, $pCO_2$ doubled and increased to 1389 µatm, which is the highest $pCO_2$ level recorded in this timeseries. The peak of $pCO_2$ was accompanied by an increase in salinity of 0.5 while temperature did not change, suggesting the influence of advection. At the beginning of August, $pCO_2$ dropped to 536 µatm and then oscillated around 600 µatm through much of August before returning to around 900 µatm for the next month. Similarly, pH decreased to 7.5 at the end of July and then oscillated around 7.85, while $\Omega_{arag}$ dropped to 0.37, and oscillated around 0.85. The steep drop and oscillation of $pCO_2$ was reflected in $NO_3$, suggesting that primary production and remineralization played a role. When $pCO_2$ and $NO_3$ decreased at the beginning of August, temperature simultaneously increased by 0.7 °C and salinity decreased by 0.12, suggesting that entrainment of shallower water masses may have played a role too. Comprehensive analyses of the factors that resulted in the 2020 differing conditions are beyond the scope of this paper, but deserve attention in a future effort."**

Line 384 If the oxygen data is already known as bad data as shown in Fig. S2, how can the author trust the oxygen changes can be used for quantitative analysis?

**We trust the relative oxygen values from the SBE63 can be used for quantitative analysis because the sensor deployed in 2017-2018 was pre- and post-calibrated, and the sensor deployed in 2018-2019 was pre-calibrated with processed data showing good agreement (relative difference of 4%) from 2018 recovery and 2018 deployment (smaller than the relative difference of 6% between the last two measurements prior to the 2018 recovery).**

Line 388 Not clear how the authors estimated TA increase with the assumption of NO3 consumption. No observation of NO3 in the 2018 spring. Need more explanation.

**In this example, we used oxygen to derive DIC, and then estimated TA with + 0.15 µmol TA per µmol DIC consumed. However, instead of carrying this analysis out in the results we now conduct a somewhat reduced thought experiment in the discussion.**

Line 404 If the water column is well-stratified, does the observation at 33 m reflect the surface mixed layer or bottom layer at this period? If it is reflecting the variation of the surface mixed layer, the CO2 air-sea exchange should be considered. If it is reflecting the variations of bottom layer, how CaCO3 mineral dissolution in the surface layer increases TA in the bottom layer?

**The water column is stratified upon sea ice break-up due to sea ice melt, so it is unlikely that there is influence of $CO_2$ air-sea gas exchange at 33 m. However, ikaite particles could penetrate the mixed layer depth and affect TA at the bottom. We moved this analysis into the discussion and discussed dissolution of ikaite as a possibility rather than a fact (L555):**

"Biogeochemistry (photosynthesis, respiration, calcification, dissolution) is the most important driver of the inorganic carbon dynamics at 33 m depth at the CEO site. The springtime relaxation events in 2018 and 2019 with relatively higher pH and $\Omega_{arag}$, and lower $pCO_2$, were mainly driven by biogeochemistry (Figure 10). During these events $O_2$ increased and $NO_3$ decreased, suggesting photosynthetic activity (Figure 2d,e and 3a). Near bottom photosynthetic activity by phytoplankton or sea ice algae has been observed at different locations across the Chukchi Sea (Arrigo et al., 2017; Ouyang et al., 2022; Stabeno et al., 2020; Koch et al., 2020). Sediment trap data from a CEO deployment prior to the start of this $pCO_2$ and pH time-series suggest that export of the exclusively sympagic sea ice algae *Nitzschia frigida* peaked in May and June, during snow and ice melt events (Lalande et al., 2020), further supporting the hypothesis that sea ice algae contributed to the $CO_2$ draw down. Interestingly, TA also increased significantly during these events in 2018 and 2019, which cannot be solely aattributed to organic matter production. Specifically, TA increased by 23 umol $kg^{-1}$ in 2019 (Figure 11a). However, with an observed $NO_3$ increase of 7.6 umol $kg^{-1}$, we would expect an increase of TA by 7.6 umol kg-1. This is assuming that $NO_3$ is the primary source of nitrogen during organic matter formation, and that assimilation of 1 umol of $NO_3$ leads to an increase of TA of 1 umol (Wolf-Gladrow et al., 2007). The TA increase of 23 umol $kg^{-1}$ is therefore larger than expected from organic matter formation alone and is likely due to $CaCO_3$ mineral dissolution. While direct evidence is missing, the strong TA increase suggests that $CaCO_3$ mineral dissolution during sea ice break up also plays an important role at the CEO site. As observed in other Arctic areas, it is possible that ikaite crystals that were trapped in the ice matrix dissolved in the water column when sea ice melted (Rysgaard et al., 2012, 2007). "

Line 407 Needs to define "fall" in this study

**We now define it (L382): In the following we will focus on spikes of high pH and $\Omega_{arag}$ and low $pCO_2$ that occur in spring (May-June) and fall (September-December); we define these spikes as relaxation events (see discussion for justification of term).**

…drops in pCO2 ?

**Was revised accordingly.**

Line 431 There is no figure (Figure 7) showing changes in Ωarag

**We added (Figures 3 and 7).**

Line 458 DIC changes or nDIC change?

**This comment is no longer applicable.**

Line 491 NO3 decreased below 10 µmol kg-1 shown in Fig. 2f

**We disagree. The figure shows that it is above 10 µmol kg$^{-1}$ in July 2020.**

Line 514 the authors need to clarify that they are comparing the acidification rate at the surface vs. in the water columns

**Thank you for pointing this out. We rewrote to (L582):**

**"Seventeen years of ship-based data from sub surface Chukchi Summer water suggests a mean pH change of -0.0047 ± 0.0026 and mean Ω$_{arag}$ change of -0.017 ± 0.009 (Qi et al., 2022b). As a comparison, an average across historic simulations from five CMIP6 models (see methods) estimates a change in pH of -0.0077 year$^{-1}$ and Ω$_{arag}$ of -0.0063 year$^{-1}$ at 33 m of the CEO site between 2002 – 2014. The historic CMIP6 simulations end in 2014 and therefore miss the last years of extreme sea ice loss. Both observations and global model-based trend estimates must be used with caution. The observations were collected during the sea ice free period (Qi et al., 2022b), and therefore do not depict an annually representative trend. Global models do not resolve important local physical, chemical, and biological meso-scale processes and therefore mask out the variability of the inorganic carbon system and effects of climate change."**

Line 515-516 I didn't see the result of the statement "The weaker trend was calculated with data starting in 1994, whereas the stronger trend used data starting eight years later." in Qi et al. (2022). Need to explain how the author concluded that.

**We rewrote this section. Please see above.**

Line 520 Please clarify if the model results were derived from the depth of the CEO.

**This was stated in the methodology section (L328): "Each simulation was used to calculate the annual trend of aragonite saturation state and pH at the closest depth and grid cell to the CEO mooring."**

**We now also clarify here (L584):**

**"As a comparison, an average across historic simulations from five CMIP6 models (see methods) estimates a change in pH of -0.0077 year$^{-1}$ and Ω$_{arag}$ of -0.0063 year$^{-1}$ at 33 m of the CEO site between 2002 – 2014."**

Line 604 …pH (e.g. Gianguzza et al., (2014).

**This was revised accordingly.**

Line 625… and lower $p$CO2, were driven by a combination

**This was revised accordingly.**

Line 627-629 It is not necessarily and exclusively to be sea ice algae, because the water column is not well-stratified in the spring, the phytoplankton is also able to grow at the subsurface or to be entrained into the subsurface.

Please check Arrigo et al., 2017 and Ouyang et al., 2022

Arrigo, K. R., Mills, M. M., van Dijken, G. L., Lowry, K. E., Pickart, R. S., & Schlitzer, R. (2017). Late spring nitrate distributions beneath the ice-covered northeastern Chukchi Shelf. Journal of Geophysical Research: Biogeosciences, 122(9), 2409-2417.

Ouyang, Z., Collins, A., Li, Y., Qi, D., Arrigo, K. R., Zhuang, Y., ... & Cai, W. J. (2022). Seasonal Water Mass Evolution and Non-Redfield Dynamics Enhance CO2 Uptake in the Chukchi Sea. Journal of Geophysical Research: Oceans, 127(8), e2021JC018326.

**We appreciate this comment, added these suggested publications, and clarified (L556):** "**The springtime relaxation events in 2018 and 2019 with relatively higher pH and $\Omega_{arag}$, and lower $p$CO$_2$, were mainly driven by biogeochemistry (Figure 10). During these events O$_2$ increased and NO$_3$ decreased, suggesting photosynthetic activity (Figure 2d, e and 3a). Near bottom photosynthetic activity by phytoplankton or sea ice algae has been observed at different locations across the Chukchi Sea (Arrigo et al., 2017; Ouyang et al., 2022; Stabeno et al., 2020; Koch et al., 2020). Sediment trap data from a CEO deployment prior to the start of this $p$CO$_2$ and pH time-series suggest that export of the exclusively sympagic sea ice algae** *Nitzschia frigida* **peaked in May and June, during snow and ice melt events (Lalande et al., 2020), further supporting the hypothesis that sea ice algae contributed to the CO$_2$ draw down.**"

Line 693 "dataset"

**Was revised accordingly.**

Line1273 I suggest enlarging Figure 2 in the y-axis scale. Since this is the most important and valuable figure in the manuscript, it should be made more readable.

Also, it is really hard to see the measured pH (grey line) in Fig. 2c. Suggest changing it to another color.

**To address this comment we split Figure 2 in to Figures 2 and 3. We also changed the grey line to more accessible red to increase readability.**

Line 1286-1287 Please enlarge the green diamond and its error bars in the figure to make it more readable.

**We revised this figure and made different color choices based on visually accessible color schemes. We enlarged the diamond and removed pH$_{SeaFET}$ on the original timestamp because it made it more difficult to interpret the figure. Figure 3 is now Figure 4.**

Supplement

Line 7-8 the format of the date should be consistent with the label in the figure.

**All formats have been changed.**

Figure S4 Suggests changing the temperature-normalized data to a more readable color.

**We don't use this figure anymore.**

Figure S4 Suggests changing the temperature-normalized data to a more readable color.

**We don't use this figure anymore.**
* * *
**RC2**
We would like to thank reviewer 2 for their constructive comments and appreciate their time to carefully read the manuscript. We agree that it is important to present new data from an Arctic coastal area and are excited that the reviewer found that this dataset presentation is within the scope of Biogeosciences. We agree with this reviewer's comments that we need to more carefully address advection and diffusion. As a result of their review and others, we realize that the paper will require major revisions before it is ready for publication. Nevertheless, we believe that once it is revised it will make an important contribution to improve our knowledge about inorganic carbon dynamics in the Chukchi Sea and will fit well into the journal Biogeosciences. Our responses are shown in **bold** below.

Review comments for manuscript: "Sea ice loss translates into major shifts in the carbonate environmental conditions in Arctic Shelf Sea" submitted to Biogeosciences by Hauri et al.

**General comments**

Time-series observations of marine biogeochemical properties in coastal regions are important for assessing environmental changes that are directly linked to peoples' livelihoods. In addition, there are few time-series stations, especially high-latitude oceans, where comprehensive biogeochemical observations have been made. Therefore, it is invaluable to present new data

in the Arctic coastal area, although there is a large amount of missing data. At this point, I believe that this manuscript can contribute to scientific progress within the scope of Biogeosciences.

However, caution should be exercised when interpreting the observed results. This is because coastal areas usually show high spatial and temporal variations in biogeochemical properties, and data from vertical one-dimensional observations cannot remove the influences of lateral advection and diffusion. The latter is particularly important in coastal areas. To interpret the observed results, it does not seem that the discussion is conducted appropriately. Overall, there was a sense of over-discussion. For these reasons, I have to judge whether the scientific quality is inadequate.

For the presentation, the resolutions of the figures are poor for the discussion of monthly variations. I recommend focusing on the relevant time series in addition to the entire time series, as shown in Fig. 2. In the introduction, I could not find the purpose of this work. It just refers presenting data. For the discussion, I had the impression that the authors made a forced interpretation of the results. One of the reasons is probably that the authors do not have a clear purpose. Based on these points, the presentation of the manuscript as a research paper is inappropriate. And, about one-third of the manuscript is used for the description of materials and methods. I recommend submission to a journal with a purpose of data presentation.

As a whole, with the current version of the manuscript, I cannot recommend publication in Biogeosciences. Judging from a lot of missing references, I think the manuscript is not completed.

Specific comments

(1) The CEO is located at the Chukchi Sea site, where water masses from the Pacific flow to the Arctic basin. Therefore, it is expected that the physical and biogeochemical properties are influenced by lateral advection and mixing. However, there were almost no words related to these influences. The authors discussed variations and changes in biogeochemical properties, mostly from the viewpoint of vertical mixing. Therefore, it is necessary to confirm that the lateral advection and mixing are negligible at the site.

**Good point. The CEO site is located between two contrasting flow fields (see the newly referenced papers by Fang et al. 2020 and Tian et al 2021). From the west, the flow originating from Bering Strait drains toward Barrow Canyon in the South. From the east, wind driven flow (even very occasional upwelling from the slope) can bring outer shelf and even basin waters to this site, although such waters are thought to mix appreciably en route so that any upwelling signal that does arrive at the mooring is small in magnitude (seen in unpublished CEO data from 2014 but not in the years since). The net result is that while there are back-and-forth flow events, the mean flow vector here is very small (Tian et al., 2021) in magnitude and the flow variance is small enough that the accumulating organic matter feeds a massive and thriving benthic hotspot – it is a depositional area for carbon so by inference**

the flow here is meek enough to allow significant export fluxes to occur (see also Grebmeier et al., 2015 and Lalande et al. 2020). We have now added text and citations to Fang and Tian to better describe the site characteristics.

weak advection. We addressed this in the manuscript with the following sentences.

(L101): The CEO site, located on the southern flank of Hanna Shoal, is a region of reduced stratification (relative to other sides of the shoal) that likely alternately feels the effects of differing flow regimes located to the west and to the east (Fang et al., 2020). Consequently, the site exhibits relatively weaker currents (Tian et al., 2021) and so is conducive to deposition of sinking organic matter that in turn feeds the local benthos (Grebmeier et al., 2015).

(L150): The CEO (71°36' N, 161°30' W, Figure 1, Hauri et al., 2018) is located along the pathway of waters flowing through Bering Strait (Fang et al., 2020) and thence from the west of Hanna Shoal toward Barrow Canyon to the south, although the wind can also drive waters from the east over the observatory site (Fang et al., 2020). From both shipboard and moored acoustic Doppler current profiler records, the south side of Hanna Shoal mean flow is characterized by a weak southward-directed current (Tian et al., 2021).

(L452): Short-term variability in $p$CO$_2$, especially in January of all three observed years, was also reflected in salinity, O$_2$ and NO$_3$ (Figure 9) and could be attributed to advection, as the CEO site is adjacent to contrasting regimes of flow and hydrographic properties (Fang et al., 2020).

(2) DIC, TA and $\Omega_{arag}$ are calculated from pCO$_2$ and pH. As pointed out by the authors, this combination leads to large error, although the error of $\Omega_{arag}$ is not so large. Therefore, it is necessary to show ranges of uncertainty as a result of calculation (propagation of errors), if authors are purposed to discuss carbon budget-like calculation. Or, it seems to be a good way to compare the calculated and observed values. But this is made only for DIC but not for TA. The authors conduct carbon budget-like discussion in some sections of the manuscript. For this, uncertainty of DIC and TA should be clarified.

We added the error envelope to figures 2 and 3, and added a section about uncertainty as part of the discussion, starting at line 500:

4.2 Uncertainty

Inherent spatial and temporal variability of the inorganic carbon parameters in the Chukchi Sea make the use of discrete water samples for evaluating sensor-based measurements difficult. Historic continuous surface measurements from the area suggest that surface $p$CO$_2$ can be as low < 250 µatm in early fall (Hauri et al., 2013), at a time of year when subsurface $p$CO$_2$ reaches its max of >800 µatm at the CEO site. This suggests a steep $p$CO$_2$ gradient of > 17 µatm per meter. High-resolution pH data from the 2017/2018 deployment suggests high temporal variability as well, further complicating the collection of

discrete water samples to adequately evaluate the sensors. The HydroC's zeroing function, in addition to our pre- and post-calibration routines that factor into the post-processing of the data, gives us confidence in the accuracy of the $pCO_2$ data, and further confidence in pH derived from $pCO_2$.

The $pH^{est}$ uncertainty of 0.0525 is likely a conservative estimate based on our validation of $pH^{est}$ (section 3.1, Table 2). Consequently, propagated uncertainties in the calculated parameters are high. As discussed in section 2.7, the pH-$pCO_2$ input pair exacerbates these larger uncertainties. Mean TA($pH^{est}$,$pCO_2$), DIC($pH^{est}$,$pCO_2$), and $\Omega_{arag}$($pH^{est}$,$pCO_2$), $\pm\ u_c$ (Orr et al., 2018) are 2173 $\pm$ 281 $\mu$mol kg$^{-1}$, 2111 $\pm$ 263 $\mu$mol kg$^{-1}$, and 0.94 $\pm$ 0.23, respectively, when input uncertainties are the standard uncertainty (Equation 1). When the input uncertainty for $pH^{est}$ is only the RMSE of 0.0161 (section 3.1), uncertainties decrease to $\pm$ 98 $\mu$mol kg$^{-1}$, $\pm$ 93 $\mu$mol kg$^{-1}$, and $\pm$ 0.09, respectively. When input uncertainties are only the random component of the input parameters (i.e. standard deviation for $pH_{SeaFET}$ and $pCO_2$ and instrument precision for T and S), TA($pH_{SeaFET}$,$pCO_2$), DIC($pH_{SeaFET}$,$pCO_2$), and $\Omega_{arag}$($pH_{SeaFET}$,$pCO_2$) $u_c$ drops to $\pm$ 38 $\mu$mol kg$^{-1}$, $\pm$ 37 $\mu$mol kg$^{-1}$, and $\pm$ 0.06, respectively. Given the above uncertainties and that we do not see significant biofouling at the CEO site, we believe that short term variability can be discussed with confidence with this dataset. In other words, wiggles in the data represent real events, despite the high uncertainty in the precise value of the calculated parameters."

(3) The pH algorithm (eq.1) is evaluated with ship-based data. But I wonder the evaluation can be related to the evaluation of pH and $pCO_2$ data obtained by sensors attached to the mooring. I think that the evaluation by ship-based data is independent to the mooring data.

The algorithm was cross-validated using the moored SeaFET timeseries for the test range (outside the training range). The fact that the independent verification of the algorithm works so well for ship-based measurements shows that the algorithm is robust across time and space. The algorithm was trained with mooring based data from the period between mid-September to mid-March and evaluated with ship-based data collected across the eastern Chukchi Sea from August and September. Figure 6 shows that the algorithm-based data is, to the most part, within 0.02 pH units of the ship-based pH, with exception of some surface data.

As noted in the manuscript, we do need another year of moored pH data to evaluate our algorithm more completely at 33 m depth at the CEO site. We look forward to doing so in the future.

(4) It is pointed (lines 681-684) out that a large gradient of $pCO_2$ exists. This implies that a slight tilt of mooring system causes an artificial variation of properties. With this condition, is it possible to detect fine carbon budget calculation? In this point also, uncertainty of the data should be clarified.

We addressed this comment on line 166:

"Pressure sensors at the top of the moorings show less than ± 1 m of excursion of the moored sensor package from its deployment mean depth in any given year, indicating that mooring blow-over or diving is not the cause of any observed large variability."

(5) At line 632, it is described that $CaCO_3$ mineral dissolution was observed. But it is very hard to find the evidence of $CaCO_3$ mineral dissolution through the manuscript. Discussion at lines 632-642 is too speculative. TA increases under the supersaturation of aragonite and calcite can be accounted for by e.g., water mixing.

**We moved the discussion around the potential drivers of the inorganic carbon dynamics into the discussion and improved this with the following edits:**

**4.3 Subsurface biogeochemical drivers of pH, $\Omega_{arag}$, and $pCO_2$**
**Inorganic carbon chemistry can be influenced by advection and vertical entrainment of different water masses, temperature, salinity, biogeochemistry, and conservative mixing with TA and DIC freshwater endmembers. Here, we followed Rheuban et al. (2019) and separated the drivers of the observed large pH, $\Omega_{arag}$, and $pCO_2$ variability to provide additional insights into our timeseries (Figure 10) using CO2SYS by altering input parameters temperature, salinity, TA, and DIC. Anomalies relative to the reference values pH($T_0$, $S_0$, $DIC_0$, $TA_0$), $\Omega_{arag}$($T_0$, $S_0$, $DIC_0$, $TA_0$), and $pCO_2$($T_0$, $S_0$, $DIC_0$, $TA_0$), were calculated using a linear Taylor series decomposition, adding up the thermodynamic effects of temperature and salinity, and the perturbations due to biogeochemistry, and conservative mixing with freshwater DIC and TA endmembers. (Rheuban et al., 2019). Reference values $T_0$, $S_0$, $DIC_0$, and $TA_0$, are the mean of the CEO timeseries. Freshwater from sea ice melt and meteoric sources (precipitation and rivers) may influence the CEO site. TA and DIC concentrations of 450 µmol kg$^{-1}$ and 400 µmol kg$^{-1}$, respectively, have been measured in Arctic sea ice (Rysgaard et al., 2007). Riverine input along the Gulf of Alaska tends to have lower TA (366 µmol kg$^{-1}$) and DIC (397 µmol kg$^{-1}$) concentrations (Stackpoole et al., 2016, 2017) than rivers draining into the Bering, Chukchi, and Beaufort Seas (TA = 1860 µmol kg$^{-1}$, DIC = 2010 µmol kg$^{-1}$, Holmes et al., 2021) all of which can influence the CEO site to some extent (Asahara et al., 2012; Jung et al., 2021). In this Taylor decomposition we used sea ice TA and DIC endmembers (Rysgaard et al., 2007) but want to emphasize that using Arctic river endmembers did not meaningfully change the results (not shown). Figure 10 shows the effects of biogeochemical processes, temperature, salinity, and conservative mixing with TA and DIC freshwater endmembers on pH, $\Omega_{arag}$, and $pCO_2$. The effects of salinity (red) and conservative mixing with TA and DIC freshwater endmembers (green) are negligible for pH, $\Omega_{arag}$, and $pCO_2$. Temperature varied between -1.7 °C during the sea ice covered months and up to 4 °C in late fall, when wind events mixed the whole water column and entrained warm and low $pCO_2$ surface waters to the instrument depth at 33 m (see section 3.2 for a more in-depth discussion of these mixing events). During this time, the increase in temperature counteracted the effect of biogeochemistry slightly and increased $pCO_2$ and decreased pH (Figure 10 a,c). Temperature did not affect $\Omega_{arag}$.**
**Biogeochemistry (photosynthesis, respiration, calcification, dissolution) is the most important driver of the inorganic carbon dynamics at 33 m depth at the CEO site. The springtime relaxation events in 2018 and 2019 with relatively higher pH and $\Omega_{arag}$, and lower**

*p*CO₂, were mainly driven by biogeochemistry (Figure 10). During these events $O_2$ increased and $NO_3$ decreased, suggesting photosynthetic activity (Figure 2d, e and 3a). Near bottom photosynthetic activity by phytoplankton or sea ice algae has been observed at different locations across the Chukchi Sea (Arrigo et al., 2017; Ouyang et al., 2022; Stabeno et al., 2020; Koch et al., 2020). Sediment trap data from a CEO deployment prior to the start of this *p*CO₂ and pH time-series suggest that export of the exclusively sympagic sea ice algae *Nitzschia frigida* peaked in May and June, during snow and ice melt events (Lalande et al., 2020), further supporting the hypothesis that sea ice algae contributed to the $CO_2$ draw down. Interestingly, TA also increased significantly during these events in 2018 and 2019, which cannot be solely attributed to organic matter production. Specifically, TA increased by 23 umol $kg^{-1}$ in 2019 (Figure 11a). However, with an observed $NO_3$ increase of 7.6 umol $kg^{-1}$, we would expect an increase of TA by 7.6 umol kg-1. This is assuming that $NO_3$ is the primary source of nitrogen during organic matter formation, and that assimilation of 1 umol of $NO_3$ leads to an increase of TA of 1 umol (Wolf-Gladrow et al., 2007). The TA increase of 23 umol $kg^{-1}$ is therefore larger than expected from organic matter formation alone and is likely due to $CaCO_3$ mineral dissolution. While direct evidence is missing, the strong TA increase suggests that $CaCO_3$ mineral dissolution during sea ice break up also plays an important role at the CEO site. As observed in other Arctic areas, it is possible that ikaite crystals that were trapped in the ice matrix dissolved in the water column when sea ice melted (Rysgaard et al., 2012, 2007).

(6) From the title of manuscript, I expected examination of climate change, which leads to "shift" of carbonate system from one condition (sea ice existence) to another (sea ice loss). However, the content of manuscript does not show shift of climatological time scale. It is however possible to see a shift of seasonal time scale. Even in this case, differences of carbonate system between conditions in the sea ice existence and those in sea ice loss should be clarified.

**We revised the title to:" Insights into carbonate environmental conditions in the Chukchi Sea"**

(7) Salinity influence is examined in the manuscript (section 2.8) but from the viewpoint of river water alone. Influence of sea-ice melting water should be also examined.

**We revised the section about influence of freshwater on inorganic carbon chemistry, addressing different freshwater sources, including sea ice melt water (L 527):**

**"Inorganic carbon chemistry can be influenced by advection and vertical entrainment of different water masses, temperature, salinity, biogeochemistry, and conservative mixing with TA and DIC freshwater endmembers. Here, we followed Rheuban et al. (2019) and separated the drivers of the observed large pH, $\Omega_{arag}$, and *p*CO₂ variability to provide additional insights into our timeseries (Figure 10) using CO2SYS by altering input parameters temperature, salinity, TA, and DIC. Anomalies relative to the reference values pH($T_0$, $S_0$, $DIC_0$, $TA_0$), $\Omega_{arag}$($T_0$, $S_0$, $DIC_0$, $TA_0$), and *p*CO₂($T_0$, $S_0$, $DIC_0$, $TA_0$), were calculated using a linear Taylor series decomposition, adding up the thermodynamic effects of temperature and salinity, and the**

perturbations due to biogeochemistry, and conservative mixing with freshwater DIC and TA endmembers. (Rheuban et al., 2019). Reference values $T_0$, $S_0$, $DIC_0$, and $TA_0$, are the mean of the CEO timeseries. Freshwater from sea ice melt and meteoric sources (precipitation and rivers) may influence the CEO site. TA and DIC concentrations of 450 µmol $kg^{-1}$ and 400 µmol $kg^{-1}$, respectively, have been measured in Arctic sea ice (Rysgaard et al., 2007). Riverine input along the Gulf of Alaska tends to have lower TA (366 µmol $kg^{-1}$) and DIC (397 µmol $kg^{-1}$) concentrations (Stackpoole et al., 2016, 2017) than rivers draining into the Bering, Chukchi, and Beaufort Seas (TA = 1860 µmol $kg^{-1}$, DIC = 2010 µmol $kg^{-1}$, Holmes et al., 2021) all of which can influence the CEO site to some extent (Asahara et al., 2012; Jung et al., 2021). In this Taylor decomposition we used sea ice TA and DIC endmembers (Rysgaard et al., 2007) but want to emphasize that using Arctic river endmembers did not meaningfully change the results (not shown). Figure 10 shows the effects of biogeochemical processes, temperature, salinity, and conservative mixing with TA and DIC freshwater endmembers on pH, $\Omega_{arag}$, and $pCO_2$. The effects of salinity (red) and conservative mixing with TA and DIC freshwater endmembers (green) are negligible for pH, $\Omega_{arag}$, and $pCO_2$."**

(8) In marine $CO_2$ community, it is required to indicate that pH is measured at which water temperature. Please describe that the pH is at in situ temperature.

**We clarified in section 2.7 by adding the following sentence (L220): "pH is reported in total scale for the entirety of this paper."**

As a whole, discussion on processes based on water mixing are acceptable, although examination of horizontal mixing should be added. But discussion on processes based on biogeochemical changes are too speculative.

**We addressed these points by conducting a Taylor decomposition and by moving the analysis around drivers into the discussion as suggested by the reviewer (see above).**

Technical corrections:

In this manuscript, the observed results and the interpretation/speculation based on the result are written together in the section of results (section 3). This may lead readers and authors themselves to misunderstanding. For example, bloom and $CaCO_3$ dissolution are described in the abstract. But these are not observations but speculation. I think that there are no problems to describe them in the main text, but description in the abstract makes readers misunderstand as if they were observed. I recommend the authors rewriting section 3, separating observed results and speculation.

**Thank you for this suggestion. We revised the results accordingly, only stating the direct findings and moved the discussion around drivers that were not directly measured into the discussion.**

Through the manuscript, "$CO_2$-deplete" or "$CO_2$ depleted" is repeatedly used. It is very rare that $CO_2$ is depleted, being different from nutrients. "$CO_2$ reduced" is better.

**Replaced "depleted" with "low" throughout the manuscript.**

Lines 56-57: If "National Snow and Ice Data Center" is a reference, add published year. Not found in the reference.

**This and all other references** were **redone and carefully checked.**

Line 69: "CaCO3", "$CaCO_3$"

**The sub and superscript errors were fixed throughout the manuscript.**

Line 70: Bednarsek et al. 2021 is cited here. But it is difficult to find the reference because of incomplete information of the reference at lines 762-764.

**We had issues with unlinking the word doc from Zotero after our final read. We apologize for these inaccuracies. All references were fixed. Lesson learned for the future.**

"CO32-", "$CO_3^{2-}$"

**Was revised accordingly.**

Line 79: Bates, 2015; Bates et al., 2009; Pipko et al., 2002; Mathis and Questel, 2013 are not found in the reference.

**Were added.**

Line 80: "CO2", "$CO_2$"

**Was revised accordingly.**

Line 81: Bates, 2015; Bates et al., 2009 are not found in the reference.

**Were added.**

Line 83: Bates et al., 2009 is not found in the reference.

**Was added.**

"pCO2", "$pCO_2$"

**Was revised accordingly.**

Lines 85-86: Mathis and Questel, 2013; Pipko et al., 2002; Bates, 2015 are not found the reference.

**Were added.**

Line 87: "pCO2", "$p$CO$_2$"

**Was revised accordingly.**

Line 88: "pCO2", "$p$CO$_2$"

**Was revised accordingly.**

Lines 88-89: Hauri et al., 2013 is not found in the reference.

**Was added.**

Line 89: Yamamoto-Kawai et al. (2016) is not found in the reference.

**Was added.**

Line 95: "CEO", "The Chukchi Ecosystem Observatory (CEO)"

**Term is now introduced in introduction (L 94) and then again in the subtitle on L 147).**

Line 96: Grebmeier et al., 2015; Moore et al., 2000 are not found in the reference.

**Were added.**

Line 98: Grebmeier et al., 2015; Blanchard et al., 2013 are not found in the reference.

**Were added.**

Line 101: Moore et al. 2022 is not found the reference.

**Was added.**

Line 111: IPCC 2022 is not found in the reference.

**Following another reviewer's comment we are now citing primary literature.**

Line 123: IPCC 2022 is not found in the reference.

**Following another reviewer's comment we are now citing primary literature.**

Line 141: "The Chukchi Ecosystem Observatory (CEO)", "The CEO"

**Was revised accordingly.**

Line 172: "quadrature", "quadruple"?

**"To add in quadrature" is a mathematical description for equation 7 in Orr et al., (2018) and has been clarified further using Equation 1.**

**Orr, J. C., Epitalon, J. M., Dickson, A. G., and Gattuso, J. P.: Routine uncertainty propagation for the marine carbon dioxide system, Mar. Chem., 207, 84–107, https://doi.org/10.1016/j.marchem.2018.10.006, 2018.**

Lines 196 and 199: "$pH_{ext}$", "$pH_{SeaFET}$", define terms at the first appearance.

**Both terms are now defined at their first appearance, thank you for pointing this out.**

**We revised the text (L208): $pH_{SeaFET}$ data were excluded during a 14-day conditioning period after the deployment and were processed with post-calibration corrected temperature and salinity from the SBE37 following** Bresnahan et al. (2014) **using voltage from the external electrode ($V_{ext}$), and $pH_{Vext}$ (pH calculated from the external electrode of the SeaFET) from an extended period of low variability (18 February 2018).**

Line 204: "quadrature", "quadruple"?

**Quadrature is correct. See above.**

Line 219: "**2.5 CTD and Oxygen**", it is better to remove the description written at lines 190-193 and summarize it here.

**We followed this suggestion and are now describing CTD and oxygen in the same section (L234):**

**"2.5 CTD and Oxygen**
**Two CTDs were deployed on the CEO mooring near the HydroC $CO_2$ depth. The main pumped Sea-Bird SeaCAT (SBE16) has been deployed on the CEO mooring around 33 m depth since 2014. A pumped SBE43 oxygen sensor was deployed with the SBE16 during the 2015-2016, 2017-2018, and 2019-2020 deployments but only data returns from the 2017-2018 deployment is discussed briefly in this manuscript (Figure S2).**
**The other pumped CTD was a Sea-Bird MicroCAT (SBE37-SMP-ODO), which was integrated with an optical dissolved oxygen sensor (SBE63; Figure S2), and the SeaFET pH sensor within the SeapHOx instrument. The SeapHOx was deployed in fall 2016, 2017, and 2018. The SBE37-SMP-ODO did not record any CTD or oxygen data during the 2016**

deployment and only recorded CTD and oxygen data between August and November 3 in 2018 due to battery failure.

Processing of these data included temperature and conductivity correction using pre- and post-calibration data following Sea-Bird Application Note 31 and oxygen correction using pre- and post-calibration data following Sea-Bird Module 28. Oxygen was converted from ml/l to µmol/kg following Bittig et al. (2018). Density and practical salinity were calculated using the TEOS-10 GSW Oceanographic Toolbox (McDougall and Baker, 2011).

Differences between the two oxygen sensors (SBE43 and SBE63) of approximately 145 to 265 µmol/kg were observed over the 2017-2018 deployment, and both moored sensors had varying offsets compared to nearby casts (Figure S2).  Therefore, only relative oxygen values from the freshly calibrated SBE63 are discussed in this paper.

The freeze-up detection mooring (Figure 6) consisted of four Sea-Bird SBE 37 inductive modem CTD sensors that transmitted in real time hourly temperature, salinity, and pressure data via the surface float from four subsurface depths (8, 20, 30, and 40 m; Hauri et al., 2018)."

Line 220: What is the CEO-2? Different from CEO?

**Not relevant here, was removed.**

"morning", "mooring"

**Was revised accordingly.**

Line 235: What do you mean "relative oxygen values from the pumped SBE63"? In the figures, absolute values seem to be used.

**We revised the figure showing relative oxygen values to the mean (now figure 3).**

Line 257: In the caption of Fig. 3, explanation of dots is not found. Indicate which displays pH$_{SeaFET}$.

**The figure was revised, also following other comments. The figure caption now reads: "Figure 4. HydroC $p$CO$_2$ and pH highlighting mirrored trend from mid-August 2017 to beginning of August 2018. Measured pH (pH$_{SeaFET}$, red dots) is interpolated onto the HydroC $p$CO$_2$ timestamp (blue), and pH$^{est}$ is shown as the solid black line. The dashed box shows the time period over which pH$^{est}$ was trained. The yellow faced diamond with error bars show reference pH$^{disc}_{calc}$ $\pm$ u$_c$ (Cross et al., 2020a; Orr et al., 2018)."**

Line 258: No explanation of black dots is found in the caption of Fig. 3. Instead, black circles are explained.

**The figure itself and the caption were revised - see above.**

Line 261: Define "pH$_{SeaFET}$" clearly.

**This was done on line 205: "Unfortunately, measured pH (pH$_{SeaFET}$) from the 2016 and 2018 SeapHOx deployments were unusable due to high levels of noise in both the internal and external electrodes. In short, we only have usable pH data between August 2017 and August 2018."**

Line 283: "quadrature", "quadruple"?

**Quadrature is correct.**

Line 290: "to be zero", "to be negligible" is better because there observed ~10-20 umol/kg NO$_3$ is observed.

**We revised to (L289): "Due to a lack of data, nutrient concentrations (Si, PO$_4$, NH$_4$, H$_2$S) were assumed to be negligible in the CO2SYS calculations (e.g. deGrandpre et al., 2019; Vergara-Jara et al., 2019; Islam et al., 2017)."**

Line 302: Show which temperature is used for pH, at in-situ temperature or a constant temperature (e.g., 25°C).

**We clarified (L220): "pH is reported in total scale and at *in situ* temperature for the entirety of this paper."**

Line 306: "(Sulpis et al., 2020)", "Sulpis et al. (2020)"

**We corrected all references and would like to apologize again.**

Lines 306-307: "(Lueker et al., 2000), "Lueker et al. (2000)"

**This was corrected.**

Line 308: "(Lueker et al., 2000), "Lueker et al. (2000)"

**This was corrected.**

Line 311: "DIC(pH$^{est}$,$p$CO$_2$)", "DIC calculated from pH$^{est}$ and $p$CO$_2$"

   **This was implemented on line 311: "Furthermore, the difference between DIC calculated from pH$^{est}$ and $p$CO$_2$ and discrete samples interpolated to moored instrument depth ranged from 266 to -195 µmol/kg using the k1 k2 of Sulpis et al. (2020), compared to -38 to -7 µmol/kg using Lueker et al. (2000)."**

Lines 336-338: The comparison does not deny influences of freshwater. It just shows that the large changes cannot be accounted for by freshwater alone.
**This section was removed. The impacts of freshwater are now described in the discussion. See above.**

Line 338: "(S3)", "(Figure S3)"?

**Figure S3 was removed because freshwater is now discussed based on figure 9.**

Line 345: "(S4)", "(Figure S4)"? Figure S4 does not display the average temperature.

**We removed figure S4 since the influence of temperature on pCO2 is now being discussed based on figure 9 (L549):**

**"Temperature varied between -1.7 °C during the sea ice covered months and up to 4 °C in late fall, when wind events mixed the whole water column and entrained warm and low $pCO_2$ surface waters to the instrument depth at 33 m (see section 3.2 for a more in-depth discussion of these mixing events). During this time, the increase in temperature counteracted the effect of biogeochemistry slightly and increased $pCO_2$ and decreased pH (Figure 10 a,c). Temperature did not affect $\Omega_{arag}$."**

Line 354: DiGirolamo et al. (2022) is not found in the reference. 2013?
**All references have been updated and checked.**

Line 361: Sivers et al. 2018 and Horowitz et al. (2018) are not found in the reference.

**These were added.**

Line 362: Boucher et al. (2020) and Seferian (2019) are not found in the reference.

**These were added.**

Lines 381-383: If sea-ice melting are associated with the TA and DIC variations, both should be decreased. As $NO_3$ also seems to be reduced, partially effects of biological activity?
**This section was completely rewritten and discussion around drivers was moved to section 4.3.**

Line 384: Increased from what?

**Relative $O_2$ changes were used.**

Line 387: Again, decreased from what?

**Absolute values do not seem relevant in this paragraph and would make the readability more difficult.**

Line 391: From Fig. 2, $\Omega_{arag}$ shows > 1.0 in the corresponding period. Can we expect dissolution of $CaCO_3$? It may be possible that it occurred as a result of dissolution. But for the explanation, time-lag is necessary between the phenomena.

**We moved the discussion around dissolution to section 4.3 (L555):**

**Biogeochemistry (photosynthesis, respiration, calcification, dissolution) is the most important driver of the inorganic carbon dynamics at 33 m depth at the CEO site. The springtime relaxation events in 2018 and 2019 with relatively higher pH and $\Omega_{arag}$, and lower $pCO_2$, were mainly driven by biogeochemistry (Figure 10). During these events $O_2$ increased and $NO_3$ decreased, suggesting photosynthetic activity (Figure 2d, e and 3a). Near bottom photosynthetic activity by phytoplankton or sea ice algae has been observed at different locations across the Chukchi Sea (Arrigo et al., 2017; Ouyang et al., 2022; Stabeno et al., 2020; Koch et al., 2020). Sediment trap data from a CEO deployment prior to the start of this $pCO_2$ and pH time-series suggest that export of the exclusively sympagic sea ice algae *Nitzschia frigida* peaked in May and June, during snow and ice melt events (Lalande et al., 2020), further supporting the hypothesis that sea ice algae contributed to the $CO_2$ draw down. Interestingly, TA also increased significantly during these events in 2018 and 2019, which cannot be solely attributed to organic matter production. Specifically, TA increased by 23 umol $kg^{-1}$ in 2019 (Figure 11a). However, with an observed $NO_3$ increase of 7.6 umol $kg^{-1}$, we would expect an increase of TA by 7.6 umol kg-1. This is assuming that $NO_3$ is the primary source of nitrogen during organic matter formation, and that assimilation of 1 umol of $NO_3$ leads to an increase of TA of 1 umol (Wolf-Gladrow et al., 2007). The TA increase of 23 umol $kg^{-1}$ is therefore larger than expected from organic matter formation alone and is likely due to $CaCO_3$ mineral dissolution. While direct evidence is missing, the strong TA increase suggests that $CaCO_3$ mineral dissolution during sea ice break up also plays an important role at the CEO site. As observed in other Arctic areas, it is possible that ikaite crystals that were trapped in the ice matrix dissolved in the water column when sea ice melted (Rysgaard et al., 2012, 2007).**

Lines 415-418: This part seems to insist that less dense water goes down to the deeper layer. I recommend discussion from the viewpoint of horizontal water transport.

**Correct, the entrainment of less dense water to deeper depths is shown in Figure 7.**

Line 448: "Bottom waters", Not waters at 33m?

**We revised to (L436):" Waters at 33 m of the CEO site were…"**

Lines 457-459: Is TA value of 4 umol $kg^{-1}$ significant?

**This is no longer applicable.**

Lines 481-484. Figure 8 displays changes of TA/DIC ratio. What does it mean and how can we realize relationship with organic matter remineralization?

**We are no longer using the TA/DIC ratio, and instead discuss drivers in terms of perturbations to the system, see section 4.3. Though the decomposition analysis does rely on TA(pH$^{est}$,pCO$_2$) and DIC(pH$^{est}$,pCO$_2$), we believe this is a valid tool for the discussion – see section 4.2 for a deeper discussion around uncertainty.**

Lines 510-528: The discussion made here is related to long-term changes of acidification. It is not appropriate to short-term changes of carbonate properties obtained in the present study.

**Yes, we are only presenting a few years of data. However, we think that it is important to talk about seasonal to interannual variability as found in our study in the context of long-term change.**

Line 611: Boyd et al. (2018) is not found in the reference.

**This citation was added.**

Lines 628-629: Stabeno et al. (2020) and Koch et al. (2020) are not found in the reference.

**Citations were added.**

Line 629: "pCO2", pCO$_2$"

**This was fixed.**

Line 632: CaCO$_3$ mineral dissolution is not observed. That is just a speculation from TA changes.

Lines 632-633: Are there any evidences that show or suggest ikaite crystal dissolution? At least, citation of previous studies is necessary.

**To address both above comments we rewrote the section about drivers and conducted a Taylor decomposition following Rheuban et al., 2019. Please see the new section (L526) "4.3 Near-bottom biogeochemical drivers of pH, Ω$_{arag}$, and $p$CO$_2$" and figures 9 and 10.**

Lines 645-646: In section 3.3, no evidence is presented, only possibility.

**We removed the discussion about denitrification.**

Lines 646-647: Increase in density does not always leads to resuspension. Data from e.g., turbidity meter are necessary.

**N/A since we removed the section on denitrification.**

Line 662: The weather quality goal of is not +/-0.003 but 0.02.

**This was corrected.**

Line 663: From Figs. 3 and 4c, it is hard to see overestimation of 0.0008. It is too good.

**Thank you, we agree and have deleted the reference to Figures 3 and 4c.**

Line 682: Hauri et al. (2013) is not found in the reference.

**Was added.**

Lines 762-764: There are no pieces of information of journal.

**This was corrected.**

Lines 893-895: There are no pieces of information of journal.

**This was corrected.**

Line 852: Insert one line before this line.

**This was corrected.**

Line 1065: "pCO2", "pCO$_2$"

**This was corrected.**

Lines 1085-1087: There are no pieces of information of journal.

**This was corrected.**

Line 1272: DiGirolamo et al., 2022 is not found in the reference.

**This was corrected.**

Line 1273: It is very hard to distinguish between black and gray on the figure.

**This figure was revised with different colors and split into two figures (now Figures 2 and 3).**

Lines 1334-1336: The top figure is a figure for O$_2$, but no explanation is given. What dashed lines indicate?

**This figure was revised, and the figure caption was adjusted accordingly.**

Table 1: Table 1 needs to be reworked. It is an unfriendly table for readers. Parameters such as $NO_3$, $pCO_2$, CTD, are placed in rows of the first column.

**This table was adjusted.**

| Deployment | Latitude | Longitude | SUNA $NO_3$ | HydroC CO2 $pCO_2$ | SBE16 CTD+ | SBE37 CTD | SeaFET pH | SBE63 $O_2$ |
|---|---|---|---|---|---|---|---|---|
| 2016-2017 | 71.5996 | -161.5184 | 1 h | 12 h (300/5 min)* | 1 h | - | - | - |
| 2017-2018 | 71.5997 | -161.5189 | 1 h | 12 h (5/5 min) | 2 h | 2 h | 2 h (30/5 min) | 2 h |
| 2018-2019 | 71.5999 | -161.5281 | 1 h | 24 h (5/5 min) | 1 h | 2 h* | - | 2 h* |
| 2019-2020 | 71.5997 | -161.5275 | 1 h | 12 h (5/5 min) | 2 h | - | - | - |
| * indicates the sensor did not return data over the whole year due to battery failure
CTD+ indicates ancillary data was available with the SBE16 file (e.g., chlorophyll fluorescence) | | | | | | | | |

Table 2: This table also needs to be reworked.

**This table seems readable as is.**
* * *
Hi Lauren,

We would like to thank you for reading our manuscript carefully and providing us with comments. We are excited to hear that you enjoyed reading the paper. As a result of the reviews provided by you and others, we realize that the paper will require major revisions before it can be published. Like you, we think that this will be an important contribution to improve our knowledge about inorganic carbon dynamics in the Chukchi Sea and believe that once revised, it will fit well into the journal Biogeosciences. You will find our responses **in bold** below. Thank you for pointing out the subscript errors and other typos. We will correct all errors in our revised version.

Hi! Great paper; I was really excited to see more information about this topic. Here are some notes.

L69-71,80-83,87-88,629: fix sub- and super-scripts

**Done.**

L99: italicize species name

**Done.**

L116: delete "and variability"

**Done.**

L111,123: can you cite primary literature instead of IPCC?

**We now cite Doney et al., (2020) to substitute IPCC on L116 and Breitberg et al., (2015) and Thomsen et al., (2013) for IPCC on L 127. We also deleted: "As ocean acidification imposes extra energy costs to most marine organisms, its effects can be amplified under food limitations" to simplify.**

L156: delete parentheses around reference since you're directly talking about it
**Done.**

L290: silicate can be quite high in this region, affecting your TA calculation. Also, the assumption of zero nutrient concentrations might work for surface water, but not near-bottom, especially as you observed the results of respiration in your pCO2 and pH, that should translate to increased nutrients as well.

**Our dataset does not include silicate or phosphate concentrations. As a substitute we used nutrient data from the Arctic Monitoring and Forecasting Center and redid the CO2sys calculations of $\Omega_{arag}$. Using nutrients in these calculations only resulted in a difference < 10^-6. We therefore decided to follow DeGranpre et al., 2019 who also assumed nutrients to be negligible in the CO2SYS calculations.**

 **We added the following sentence:**

**L289: "Due to a lack of data, nutrient concentrations (Si, $PO_4$, $NH_4$, $H_2S$) were assumed to be negligible in the CO2SYS calculations (e.g. deGrandpre et al., 2019; Vergara-Jara et al., 2019; Islam et al., 2017)."**

L293: I agree. Can you be more specific about the uncertainty that the pH-pCO2 pair can lead to? Especially because you're using pCO2 and pH calculated from it, so really just pCO2. What range or at least order of magnitude of error do you expect this to lead to in your reported TA, DIC, and sat arag? Additionally include error propagation from CO2SYS calculations (should be available in most new versions). To that end I think you need error envelopes around your data lines in Figure 2.

We added a new section L500:

**"4.2 Uncertainty**

**Inherent spatial and temporal variability of the inorganic carbon parameters in the Chukchi Sea make the use of discrete water samples for evaluating sensor-based measurements difficult. Historic continuous surface measurements from the area suggest that surface $pCO_2$ can be as low < 250 µatm in early fall (Hauri et al., 2013), at a time of year when subsurface $pCO_2$ reaches its max of >800 µatm at the CEO site. This suggests a steep $pCO_2$ gradient of > 17 µatm per meter. High-resolution pH data from the 2017/2018 deployment suggests high temporal variability as well, further complicating the collection of discrete water samples to adequately evaluate the sensors. The HydroC's zeroing function, in addition to our pre- and post-calibration routines that factor into the post-processing of the data, gives us confidence in the accuracy of the $pCO_2$ data, and further confidence in pH derived from $pCO_2$.**

**The pH$^{est}$ uncertainty of 0.0525 is likely a conservative estimate based on our validation of pH$^{est}$ (section 3.1, Table 2). Consequently, propagated uncertainties in the calculated parameters are high. As discussed in section 2.7, the pH-$pCO_2$ input pair exacerbates these larger uncertainties. Mean TA(pH$^{est}$,$pCO_2$), DIC(pH$^{est}$,$pCO_2$), and $\Omega_{arag}$(pH$^{est}$,$pCO_2$), $\pm$ u$_c$ (Orr et al., 2018) are 2173 $\pm$ 281 µmol kg$^{-1}$, 2111 $\pm$ 263 µmol kg$^{-1}$, and 0.94 $\pm$ 0.23, respectively, when input uncertainties are the standard uncertainty (Equation 1). When the input uncertainty for pH$^{est}$ is only the RMSE of 0.0161 (section 3.1), uncertainties decrease to $\pm$ 98 µmol kg$^{-1}$, $\pm$ 93 µmol kg$^{-1}$, and $\pm$ 0.09, respectively. When input uncertainties are only the random component of the input parameters (i.e. standard deviation for pH$_{SeaFET}$ and $pCO_2$ and instrument precision for T and S), TA(pH$_{SeaFET}$,$pCO_2$), DIC(pH$_{SeaFET}$,$pCO_2$), and $\Omega_{arag}$(pH$_{SeaFET}$,$pCO_2$) u$_c$ drops to $\pm$ 38 µmol kg$^{-1}$, $\pm$ 37 µmol kg$^{-1}$, and $\pm$ 0.06, respectively. Given the above uncertainties and that we do not see significant biofouling at the CEO site, we believe that short term variability can be discussed with confidence with this dataset. In other words, wiggles in the data represent real events, despite the high uncertainty in the precise value of the calculated parameters."**

**We also added uncertainty envelops to Figures 2 and 3.**

L374: correct to "drivers"

**Done.**

L385: here and all the times after, use Greek letter μ instead of u

**Done.**

L395: In this whole discussion I think it's important to include your uncertainties in observations of DIC or TA increase (I suspect could be quite high), as that will inform how disparate the Redfield ratio calculations actually are from your measurements.

**We added a new section about uncertainty, starting on L500. Figures 2 and 3 now also include uncertainty envelopes.**

L409: change "we're attributing" to "we attribute"

**Done.**

L421: delete "to" leaving "temperature increase"

**Done.**

L437: Can you discuss this in the context of nutrient data as well? Is it possible that some amount of pCO2 decrease is also attributable to mixing stimulating phytoplankton blooms?

**Mixing plays a major role in the decrease of $p$CO$_2$ since salinity and temperature changes happened simultaneously. at the same time as $p$CO$_2$. With the data we have we are not able to distinguish between a stimulated bloom because of mixing or mixing down low $p$CO$_2$ waters. However, the evidence we have points to mixing low surface $p$CO$_2$ water to the CEO site.**

L466: Again I think error discussion is important here (as in write 85.5 +/- X umol/kg), the error could easily be on the order of these discrepancies

**We added uncertainties to the corresponding figures and discussed it more broadly in section 3.3.**

L554: Correct 'maybe' to 'may'

**Done.**

L612: I find this paragraph too speculative (off the back of the paragraph discussing the uncertainty about the carbonate chemistry impacts on organisms here) for the serious claims it brings up

**We rephrased paragraph to (L679): "Indigenous communities are at the forefront of the changing Arctic, including changes in accessibility, availability, and condition of traditional marine foods (Buschman and Sudlovenick, 2022; Hauser et al., 2021). Several marine species are critical to the food and cultural security of coastal Inupiat who have thrived in Arctic Alaska for millenia. While it is not possible to resolve the consequences of the seasonal and interannual variations in carbonate chemistry documented in this manuscript without a proper sensitivity evaluation, the seasonally low pH conditions have the potential to impact organisms like bivalves in a foraging hotspot for walrus (Jay et al., 2012; Kuletz et al., 2015). Walrus, as well as their bivalve stomach contents, are important nutritional, spiritual, and cultural components, raising concerns for food security in the context of ecosystem shifts associated with the variability and multiplicity of climate impacts within the region (ICC, 2015)."**

L625: change 'was' to were' (subject is 'events')

**Done.**

L630: italicize species name

**Done.**

L650-653: again a bold claim for how speculative it is. Also see Mordy et al. 2020 Deep Sea Res II, they have nitrate data from a very similar region (looks like you referenced them but not for your discussion here)

**We do not discuss denitrification any longer.**

L663: overestimates pH "by" …

**Done.**

L666: air-sea gas…

**Done.**

L679: I think this is an important discussion and highlights the necessity for time-series monitoring like this, especially since most of the long-term changes in pH and sat arag are

assumed from some discrete datasets that are biased to fall, which you identify as the high-pCO2 time period

**Agreed.**

Overall, I find this paper interesting and high temporal resolution measurements are super important due to the historical paucity of data, as we try to parse future changes due to climate change. The main thing to change here is the lack of transparency regarding uncertainty in carbonate chemistry parameters, especially since most of them are calculated and not directly measured. I really think that could alter some of your discussion and implications. I also wonder if the discussion couldn't focus more on physical processes than on biological impacts, since by your own admission those are relatively unclear for organisms in the Chukchi Sea?

Thanks very much for an interesting read!

---

## Author Response (AR2)

**Dear Reviewer 1,**

**We would like to thank you for taking your valuable time to read and re-evaluate our manuscript once again. We really appreciate all you have contributed to improving this work. Thank you.**

Summary:
Thank you for your hard work to respond both reviewers' comments. I find that the revised version of the manuscript has been greatly improved and most of the concerns I brought up in the previous round review have been addressed. Although I recommend the publication of the current version of the manuscript, I still have some minor comments for the authors to consider.

Minor comments:

Line 36: …wind-driven mixing event…
**Done.**

Line 383: How do you define the "relaxation events"? When $\Omega arag$ is >1 and lasts for a few days during summertime?
**We did not use an absolute value as a definition. As stated in the manuscript, we looked at that exhibited relatively higher pH and $\Omega_{arag}$ and lower $p$CO$_2$ compared to the overall mean.**

Line 409: Figure 8a does not present temperature data
**The red line does represent temperature data (see right y-axis).**

Line 420: Figure 3? and 8
**Replaced Figure 8 by Figure 2b and c, 3a and 8**

Line 450: Figure 9?
**No, Figure 9 focuses on winter months. We added instead:**
**(Figure 2d and 3b-d)**

Line 451: 1) O2 doesn't decrease much over the ice-covered period. 2) The O2 concentration data in Fig.9 from SBE63 is not accurate (too high) and misleading since only the relative change in O2 was considered.
**This corresponds with what we stated:**
        **At the same time, NO$_3$ slowly increased and O$_2$ decreased, which points to slow organic matter remineralization (Figure 9). Short-term variability in $p$CO$_2$, especially in January of all three observed years, was also reflected in salinity, O$_2$ and NO$_3$ (Figure 9) and**

**could be attributed to advection, as the CEO site is adjacent to contrasting regimes of flow and hydrographic properties (Fang et al., 2020).**

Line 489: … the western Beaufort Seas…
**Done.**

Line 545: …endmembers (Rheuban et al., 2019).
**(Rysgaard et al., 2007) is the right citation here since we used sea ice endmembers from them.**

Line 545-558: It is not clear how the authors conduct a Taylor decomposition analysis through they referred the method to Rheuban et al., (2019). I think it would be better to briefly introduce it in the Method section.
**We feel that we gave an extensive explanation on how we applied Rheuban et al., 2019 on line 529 through 552. The Taylor expansion is explained step by step in Rheuban et al., 2019 and can be easily applied thanks to their effort to make it understandable. We therefore don't think it is necessary to additionally explain it here, especially given the other reviewer's comment on how technical our text is.**

Figure 10: The black line needs to be defined. The different colored lines are not readable.
**We revised the figure with dashed lines and switched the red line to turquoise to make it readable to people who have issues distinguishing red from green. The black line is defined now in the caption:    Contributions of changes in salinity (red), temperature (blue), biogeochemistry (pink), and freshwater mixing (green) to changes (black, relative to the mean of the timeseries) in pH, $\Omega_{arag}$, and $pCO_2$ were computed following Rheuban et al. (2019). The grey dotted line illustrates an estimated residual term.**

**Dear Reviewer 2,**

**We would like to thank you for taking your valuable time to read and re-evaluate our manuscript once again. We really appreciate all you have contributed to improving this work. Thank you.**

General comments
The authors respond well to reviewers' comments. However, there are a few points I'd like to ask the authors to review (see Specific Comments and Technical Corrections).
Despite the well-revised manuscript, I still have the impression that the manuscript is difficult

to read. This is probably due to the fact that technological investigations and scientific results appear together in one manuscript. Nearly half of the manuscript is devoted to observational techniques and data recovery. I agree that the description is important in examining scientific results, but too much. I recommend to divide the manuscript into a technological and a scientific part, the former being submitted as a technical paper. Or it may be better to summarize the technical description in an appendix. I strongly recommend that authors choose one of the above alternatives.

**We appreciate this comment. However, we believe that it is very important to keep the technical aspects of the observatory and analysis of the data combined with the scientific results.**

Specific comments
Are values of pCO2 and pH in situ or at sea level?
**We changed the sentence to: "pH is reported in total scale and at *in situ* temperature and depth for the entirety of this paper."**

Lines 83-84: "as a result of organic matter". Organic matter itself does not influence pCO2. Organic matter remineralization?
**Corrected – thank you!**

Lines 386-392: By close examination of Figs. 2 and 3, the relaxation event seems to have occurred in August or later. Or not 2020 but 2018? Check this point.
**Yes – it should have been 2018 and not 2020. This is corrected.**

Lines 568-571: I feel something contradicted in this part. The assimilation implies consumption of NO3, i.e., decrease of NO3. From the NO3 increase of 7.6 umol kg-1, is an increase of TA?
**Thank you for catching this. It should read:**
**However, with an observed NO$_3$ decrease of 7.6 umol kg$^{-1}$, we would expect an increase of TA by 7.6 umol kg-1.**

Section 4.4: The first half of the discussion in this section seems unnecessary. The progression of ocean acidification is based on model results, but because of the short observations at the CEO site, it is unreasonable to compare model results with the CEO observations. I recommend deleting this part and limiting the discussion to the relationship between longer open water seasons and the relaxation events.
**We agree and deleted the first paragraph.**

Technical corrections:
Lines 55-56: "National Show and Ice Data Center, 2017"
**Replace with (National Snow and Ice Data Center, DiGirolamo et al. (2022))**

Lines 78, 81 and 85: "Bates, 2015" is not found in the reference.
**The citation was added.**

Lines 100 and 1156-1163: "Moore et al., 2022". In the reference, there are two "Moore et al., 2022". Distinguish each other.
**The "Moore et al., 2022" sea ice paper had to be removed.**

Line 291: The published year "2017" is different from that in the reference list at line 1025.
**Thanks for pointing this out. Reference year was corrected to 2017.**

Line 313: "k1 k2", "K1* and K2*"
**Done.**

Lines 345 and 346: Is "a mean of 0.0008" same to "mean difference of 0.0008"? If so, why is it repeated?
**Agreed – we deleted the parenthesis.**

Line 356: "SKQ202014S", is this a cruise code?
**We added "cruise".**

Line 359: "was fit to pHdisc calc", "was fit to pCO2disccalc"?
**No, it is correct as written.**

Table 1: Latitude and longitude should be expressed with "N or S" and "E or W" respectively according to a conventional style.
**Done.**

Figure 3: The uncertainty envelope is not visible, especially in the alternating gray. I recommend changing the colors.
**We changed the background gray to yellow in both figures (2 and 3).**

Figure 10: I cannot see the colored lines, especially the blue, green, and gray lines. I recommend inserting symbols.
**We revised figure 10 to make the lines more visible. However, the blue, green and gray lines are stagged on top of each other and therefore hard to visualize.**